# Discovering Design Concepts for CAD Sketches

**Yuezhi Yang**[*]
The University of Hong Kong
Microsoft Research Asia
yzyang@cs.hku.hk

**Hao Pan**
Microsoft Research Asia
haopan@microsoft.com

## Abstract

Sketch design concepts are recurring patterns found in parametric CAD sketches. Though rarely explicitly formalized by the CAD designers, these concepts are implicitly used in design for modularity and regularity. In this paper, we propose a learning based approach that discovers the modular concepts by induction over raw sketches. We propose the dual implicit-explicit representation of concept structures that allows implicit detection and explicit generation, and the separation of structure generation and parameter instantiation for parameterized concept generation, to learn modular concepts by end-to-end training. We demonstrate the design concept learning on a large scale CAD sketch dataset and show its applications for design intent interpretation and auto-completion.

## 1 Introduction

Parametric CAD modeling is a standard paradigm for mechanical CAD design nowadays. In parametric modeling, CAD sketches are fundamental 2D shapes used for various 3D construction operations. As shown in Fig. 1, a CAD sketch is made of primitive geometric elements (e.g. lines, arcs, points) which are constrained by different relationships (e.g. coincident, parallel, tangent); the sketch graph of primitive elements and constraints captures design intents, and allows adaptation and reuse of designed parts by changing parameters and updating all related elements automatically [1]. Designers are therefore tasked with the meticulous design of such sketch graphs, so that the inherent high-level design intents are easy to interpret and disentangle. To this end, meta-structures (Fig. 1), which we call *sketch concepts* in this paper, capture repetitive design patterns and regulate the design process with more efficient intent construction and communication [9, 12]. Concretely, each sketch concept is a structure that encapsulates specific primitive elements and their compositional constraints, and the interactions of its internal elements with outside only go through the interface of the concept.

How to discover these modular concepts automatically from raw sketch graphs? In this paper, we cast this task as a program library induction problem by formulating a domain specific language (DSL) for sketch generation, where a sketch graph is formalized as a program, and sketch concepts are modular functions that abstract primitive elements and compose the program (Fig. 1). Discovering sketch concepts thus becomes the induction of library functions from sketch programs. While previous works address the general library induction problem via expensive combinatorial search [20, 5, 7], we present a simple end-to-end deep learning solution for sketch concepts. Specifically, we bridge the implicit and explicit representations of sketch concepts, and separate concept structure generation from parameter instantiation, so that a powerful deep network can detect and generate sketch concepts, by training with the inductive objective of reconstructing sketch with modular concepts.

We conduct experiments on large-scale sketch datasets [17]. The learned sketch concepts show that they provide modular interpretation of design sketches. The network can also be trained on incomplete input sketches and learn to auto-complete them. Comparisons with state-of-the-art approaches that

---

[*]Work done during internship at Microsoft Research Asia.

36th Conference on Neural Information Processing Systems (NeurIPS 2022).

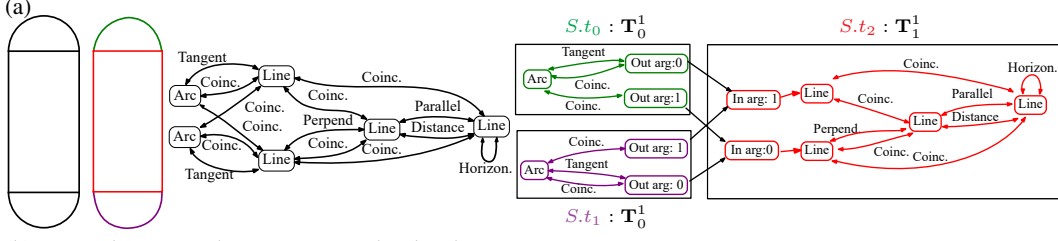

(a)

(b) Learned program that restructures the sketch:

$$\mathbf{T}_0^1 \rightarrow \lambda(\alpha_0^o, \alpha_1^o).\{t_0\text{:Arc}, t_1\text{:Tang.}, t_2, t_3\text{:Coinc.}, R=\{t_1(t_0, \alpha_0^o), t_2(t_0, \alpha_0^o), t_3(t_0, \alpha_1^o)\}\}$$

$$\mathbf{T}_1^1 \rightarrow \lambda(\alpha_0^i, \alpha_1^i).\{t_0, t_1, t_2, t_3\text{:Line}, t_4, t_5, t_6, t_7\text{:Coinc.}, t_8\text{:Perpend.}, t_9\text{:Parallel}, t_{10}\text{:Distance}, t_{11}\text{:Horizon.},$$
$$R=\{t_4(t_0, t_3), t_5(t_0, t_2), t_0(t_1, t_2), t_7(t_1, t_3), t_8(t_1, t_2), t_9(t_2, t_3), t_{10}(t_2, t_3), t_{11}(t_3), \alpha_0^i(t_1), \alpha_1^i(t_0)\}\}$$

$$S \rightarrow \{t_0, t_1\text{:}\mathbf{T}_0^1, t_2\text{:}\mathbf{T}_1^1, R=\{t_0(t_2.\alpha_1^i, t_2.\alpha_0^i), t_1(t_2.\alpha_0^i, t_2.\alpha_1^i)\}\}$$

Figure 1: **Concept learning from sketch graphs**. **(a)** In black are the raw sketch and its constraint graph, with nodes showing primitives and edges depicting constraints. Colored are the restructured sketch and its modular constraint graph, where each module box represents a concept; primitives and constraint edges are colored according to the modular concepts. **(b)** The restructured sketch graph in our DSL program representation (List 1), where the whole sketch $S$ is compactly constructed with three instances of two learned $\mathbb{L}^1$ types. We simplify notation super/sub-scripts for readability.

solve sketch graph generation through autoregressive models show that the modular sketch concepts learned by our approach enable more accurate and interpretable completion results.

To summarize, we make the following contributions in this paper:

- We formulate the task of discovering modular CAD sketch concepts as program library induction for a declarative DSL modeling sketch graphs.
- We propose a self-supervised deep learning framework that discovers modular libraries for the DSL with simple end-to-end training.
- We show the framework learns from large-scale datasets sketch concepts capturing intuitive and reusable components, and enables structured sketch interpretation and auto-completion.

## 2 Related work

**Concept discovery for CAD sketch** It is well acknowledged in the CAD design community that design intents are inherent to and implicitly encoded by the combinations of geometric primitives and constraints [12, 10]. However, there is generally no easy approach to discover the intents and make them explicit, albeit through manual design of meta-templates guided by expert knowledge [9, 10]. We propose an automatic approach to discover such intents, by formulating the intents as modular structures with self-contained references, and learning them through self-supervised inductive training with simple objectives on large raw sketch dataset. Therefore, we provide an automatic approach for discovering combinatorially complex structures through end-to-end neural network learning.

**Generative models for CAD sketch** A series of recent works [6, 24, 13, 18, 25] use autoregressive models [22] to generate CAD sketches and constraints modeled through pointer networks [23]. These works focus on learning from large datasets [17] to generate plausible layouts of geometric primitives and their constraints, which can then be fine-tuned with a constraint solver for more regular sketches. Different from these works, our aim is to discover modular structures (i.e. sketch concepts) from the concrete sketches. Therefore, our framework provides higher-level interpretation of raw sketches and more transparent auto-completion than these works (cf. Sec. 6).

**Program library induction for CAD modeling** Program library induction has been studied in the shape modeling domain [7]. General program synthesis assisted by deep learning is a research topic with increasing popularity [20, 3, 4, 19, 5]. The library induction task specifically involves combinatorial search, as has been handled by neural guided search [20, 5] or by pure stochastic sampling [7]. We instead present an end-to-end learning algorithm for sketch concept induction. In particular, based on key observations about sketch concepts, we present implicit-explicit dual representations of concept library functions, and separate the concept structure generation from parameter instantiation, to enable self-supervised training with induction objectives.

**List 1:** A domain-specific language formulating CAD sketch concepts

```
// Basic data types
```
Length, Angle, Coord, Ref
```
// L⁰ primitive types
```
Line $\rightarrow c_{start\_x}, c_{start\_y}, c_{end\_x}, c_{end\_y}$ : Coord
Circle $\rightarrow c_{center\_x}, c_{center\_y}$ : Coord, $l_{radius}$ : Length
$\cdots$
```
// L⁰ constraint types
```
Coincident $\rightarrow \lambda(r_1, r_2 : \text{Ref}).\{\}$
Parallel Distance $\rightarrow \lambda(r_1, r_2 : \text{Ref}).\{l_{dist} : \text{Length}\}$
$\cdots$
```
// L¹ composite types
```
$\mathbf{T}_i^1 \rightarrow \lambda([\alpha_k : \text{Ref}]).\{t_{i,j}^0 : \mathbf{T}_j^0 \in \mathbb{L}^0, R_{\mathbf{T}_i^1}\left([t_{i,j}^0]\cup[\alpha_k]\right)\}$
```
// Sketch decomposition
```
$S \rightarrow \{t_i^1 : \mathbf{T}_i^1 \in \mathbb{L}^1, R_S([t_i^1])\}$

# 3 CAD sketch concept formulation

To capture the notion of sketch concepts precisely, we formulate a domain specific language (DSL) (syntax given in List 1, an exhaustive list of data types given in the supplementary). In the DSL, we first define the basic data types, including *length, angle, coordinate,* and the *reference* type, where a reference binds to another reference or a primitive for modeling the constraint relationships. Second, we define the $\mathbb{L}^0$ collection of primitive and constraint types as given in raw sketches. In particular, we regard the constraints as functions whose arguments are the references to bind with primitives, e.g. a coincident constraint $c = \lambda(r_1, r_2 : \text{Ref}).\{\}$, where a function is represented in the lambda calculus style (one may refer to [14] for introductory lambda calculus formality). Some constraints have parameters other than mere references, which are treated as variables inside, e.g. parallel distance in List 1[2]. Third, we define the sketch concepts as $\mathbb{L}^1$ types composed of $\mathbb{L}^0$ types. To be specific, a composite type $\mathbf{T}_i^1 \in \mathbb{L}^1$ is a function with arguments $[\alpha_k]$ and members $t_{i,j}^0 : \mathbf{T}_j^0 \in \mathbb{L}^0$, which are connected through a composition operator $R_{\mathbf{T}_i^1} = \{p(q)|p, q \in [t_{i,j}^0]\cup[\alpha_k]\}$ that specifies how each pair of primitive elements binds together. For example, a coincident constraint $p = \lambda(r_1, r_2).\{\}$ may take a line primitive $q$ as its first argument and bind to an argument $\alpha_k$ of the composite type as its second argument, i.e. $p(q, \alpha_k) \in R_{\mathbf{T}_i^1}$; on the other hand, an argument $\alpha_k$ may bind to a primitive $q$, which is specified by $\alpha_k(q) \in R_{\mathbf{T}_i^1}$. Finally, an input sketch $S$ is restructured as a collection of composite types $t_i^1 : \mathbf{T}_i^1 \in \mathbb{L}^1$, as well as their connections specified by a corresponding composition operator $R_S$. $R_S$ records how different concepts bind through their arguments, which further transfers to $\mathbb{L}^0$ typed elements inside the concepts and translates into the raw constraint relationships of the sketch graph. Fig. 1(b) shows an example DSL program encoding sketches and concepts.

Given the explicit formulation of CAD sketches through a DSL, the discovery of sketch concepts becomes the task of learning program libraries $\mathbb{L}^1$ by induction on many sketch samples. Therefore, our task resembles shape program synthesis that aims at building modular programs for generating shapes [5, 7], and differs from works that use autoregressive language models to generate CAD sketch programs one token at a time [6, 13, 18]. In Sec. 6.2, we show that the structured learning of CAD sketches enables more robust auto-completion than unstructured language modeling.

The search of structured concepts is clearly a combinatorial problem with exponential complexity, which is intractable unless we can exploit the inherent patterns in large-scale sketch datasets. However, to enable deep learning based detection and search of structured concepts, we need to bridge the implicit deep representations and the explicit and interpretable structures, which we build through the following two key observations:

- **A concept has dual representations**: implicit and explicit. The implicit representation as embeddings in latent spaces is compatible with deep learning, while the explicit representation provides structures on which desired properties (e.g. modularity) can be imposed.

---

[2]While other works [13, 18] have skipped such constraints, we preserve them but omit generating the parameter values that can be reliably deduced from primitives. See more discussions in the supplementary.

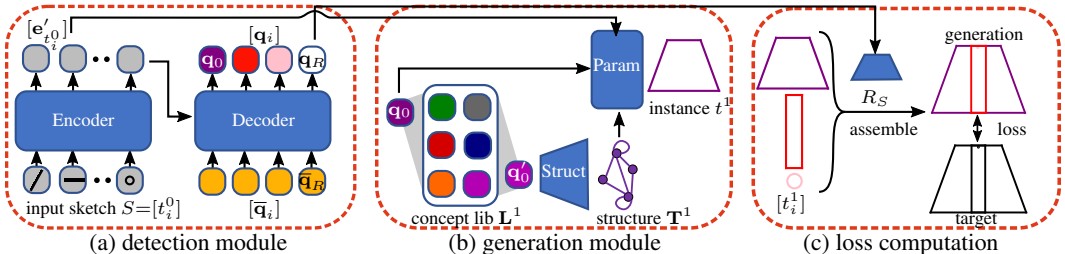

| (a) detection module | (b) generation module | (c) loss computation |

**Figure 2: Framework illustration. (a)** The detection module is a transformer network that detects from the sketch sequence $[t_i^0]$ implicitly encoded concepts $[\mathbf{q}_i]$ and their composition $\mathbf{q}_R$. **(b)** Each $\mathbf{q}$ is quantized against the concept library $\mathbf{L}^1$ to obtain prototype $\mathbf{q}'$, which is expanded by structure network into an explicit structure $\mathbf{T}^1$ and further instantiated by parameter network into $t^1$. **(c)** The collection of $[t_i^1]$ are assembled by composition operator $R_S$ generated from $\mathbf{q}_R$ to obtain the final generated sketch graph, which is compared with the input sketch for loss computation.

- **A concept is a parameterized structure**. A concept is a composite type with fixed modular structure for interpretability, but the structure is always instantiated by assigning parameters to its component primitives when the concept is found in a sketch.

## 3.1 Method overview

According to the two observations, we design an end-to-end sketch concept learning framework by self-supervised induction on sketch graphs. As shown in Fig. 2, the framework has two main steps before loss computation: a detection step that generates implicit representations of concepts making up the input sketch, and an explicit generation step that expands the implicit concepts into concrete structures on which self-supervision targets like reconstruction and modularity are applied.

Building on a state-of-the-art detection architecture [2], the detection module $D$ takes a sketch $S$ as input and detects the modular concepts within it, i.e. $\{\mathbf{q}_i\} = D(S, \{\overline{\mathbf{q}}_i\})$, where the concepts are represented implicitly as latent codes $\{\mathbf{q}_i\}$, and $\{\overline{\mathbf{q}}_i\}$ are a learnable set of concept instance queries. Notably, we apply vector quantization to the latent codes and obtain $\{\mathbf{q}_i' = \min_{\mathbf{p} \in \mathbb{L}^1} ||\mathbf{p} - \mathbf{q}_i||_2\}$, which ensures that each concept is selected from the common collection of learnable concepts $\mathbb{L}^1$ used for restructuring all sketches.

The explicit generation module is separated into two sub-steps, structure generation and parameter instantiation, which ensures that the modular concept structures are explicit and reused throughout different sketch instances. Specifically, the structure network takes each quantized concept code $\mathbf{q}_i'$ and generates its explicit form $\mathbf{T}_i^1$ in terms of primitives and constraints of $\mathbb{L}^0$ types along with the composition operator $R_{\mathbf{T}_i^1}$. Subsequently, the parameter network instantiates the concept structure by assigning parameter values to each component of $\mathbf{T}_i^1$ conditioned on $\mathbf{q}_i$ and input sketch, to obtain $t_i^1$.

The composition operator $R_S$ for combining $\{t_i^1\}$ is generated from a special latent code $\mathbf{q}_R$ transformed by $D$ from a learnable token $\overline{\mathbf{q}}_R$ appended to $\{\overline{\mathbf{q}}_i\}$.

The entire model is trained end-to-end by reconstruction and modularity objectives. In particular, we design loss functions that measure differences between the generated and groundtruth sketch graphs, in terms of both per-element attributes and pairwise references. Given our explicit modeling of encapsulated structures of the learned concepts, we can further enhance the modularity of the generation by introducing a bias loss that encourages in-concept references.

# 4 End-to-end sketch concept induction

## 4.1 Implicit concept detection

**Sketch encoding** A raw sketch $S$ can be serialized into a sequence of $\mathbb{L}^0$ primitives and constraints. Previous works have adopted slightly different schemes to encode the sequence [6, 13, 18, 24, 25]. In this paper, we build on the previous works and take a simple strategy akin to [13, 25] for input sketch encoding. Specifically, we split each $\mathbb{L}^0$ typed instance $t^0$ into several tokens: *type*, *parameter*, and a list of *references*. For each of the token category, we use a specific embedding module. For

example, parameters as scalars are quantized into finite bins before being embedded as vectors (see supplementary for the quantization details), and since there are at most five parameters for each primitive, we pack all parameter embeddings into a single code. On the other hand, each constraint reference as a primitive index is directly embedded as a code. Therefore, each token of a $\mathbb{L}^0$ typed instance is encoded as

$$\mathbf{e}_{t^0.x} = \text{enc}_{type}(t^0) + \text{enc}_{pos}(t^0.x) + \left[\text{enc}_{param}(t^0.x)|\text{enc}_{ref}(t^0.x)\right], \tag{1}$$

where $t^0.x$ iterates over the split tokens (i.e., type, parameters, references), the type embedding is shared for all tokens of the instance, the position embedding counts the token index in the whole split-tokenized sequence of $S$, and parameter or reference embeddings are applied where applicable.

**Concept detection** We build the detection network as an encoder-decoder transformer following [2]. The transformer encoder operates on the sketch encoded sequence $[\mathbf{e}_{t_i^0 \in S}]$ and produces the contextualized sequence $[\mathbf{e}'_{t_i^0 \in S}]$ through layers of self-attention and feed-forward. The transformer decoder takes a learnable set of concept queries $[\overline{\mathbf{q}}_i]$ of size $k_{qry}$ plus a special query $\overline{\mathbf{q}}_R$ for composition generation, and applies interleaved self-attention, cross-attention to $[\mathbf{e}'_{t_i^0}]$ and feed-forward layers to obtain the implicit concept codes $[\mathbf{q}_i]$ and $\mathbf{q}_R$. The concept codes are further quantized into $[\mathbf{q}'_i]$ by selecting concept prototypes from a library $\mathbf{L}^1$ implicitly encoding $\mathbb{L}^1$, before being expanded into explicit forms.

## 4.2 Explicit concept structure generation

**Concept structure expansion** Given a library code $\mathbf{q}' \in \mathbf{L}^1$ representing a type $\mathbf{T}^1 \in \mathbb{L}^1$, through an MLP we expand its explicit structure as a collection of codes $[\mathbf{t}_i^0]$ representing the $\mathbb{L}^0$ type instances $[t_i^0]$ and a matrix representing the composition $R_{\mathbf{T}^1}$ of $[t_i^0]$ and arguments (cf. List 1).

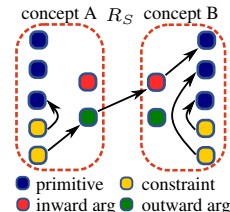

concept A $R_S$ concept B

● primitive ○ constraint
● inward arg ● outward arg

We fix the maximum number of $\mathbb{L}^0$ type instances to $k_{L^0}$ (12 by default), and split the arguments into two groups, *inward arguments* and *outward arguments*, each of maximum number $k_{arg}$ (2 by default). Each type code $\mathbf{t}_i^0$ is decoded into discrete probabilities over $\mathbb{L}^0$ with an additional probability for null type $\phi$ to indicate the emptiness of this element (cf. Sec. 5.1), by $\text{dec}_{type}(\cdot)$ as the inverse of $\text{enc}_{type}(\cdot)$ in Sec. 4.1. An inward argument only points to a primitive inside the concept structure and originates from a constraint outside, and conversely an outward argument only points to primitives outside and originates from a constraint inside the concept (see inset for illustration); the split into two groups eases composition computation, as discussed below.

The composition operator $R_{\mathbf{T}^1}$ is implemented as an assignment matrix $\mathbf{R}_{\mathbf{T}^1}$ of shape $(2k_{L^0}+k_{arg}) \times (k_{L^0}+k_{arg})$, where each row corresponds to a constraint reference or inward argument, and each column to a primitive or outward argument. The two-fold coefficient of constraint references comes from that any constraint we considered in the dataset [17] has at most two arguments. Each row is a discrete probability distribution such that $\sum_j \mathbf{R}_{\mathbf{T}^1}[i, j] = 1$, with the maximum entry signifying that the $i$-th constraint/outward argument refers to the $j$-th primitive/inward argument. We compute $\mathbf{R}_{\mathbf{T}^1}$ by first mapping the concept code $\mathbf{q}'$ to a matrix of logits in the shape of $\mathbf{R}_{\mathbf{T}^1}$, and then applying softmax transform for each row. Notably, we avoid the meaningless loops of an element referring back to itself, and inward arguments referring to outward arguments, by masking the diagonal blocks $\mathbf{R}_{\mathbf{T}^1}[2i{:}2i{+}2, i], i \in [k_{L^0}]$ and the argument block $\mathbf{R}_{\mathbf{T}^1}[2k_{L^0}{:}, k_{L^0}{:}]$ by setting their logits to $-\infty$.

**Cross-concept composition** Aside from references inside a concept, references across concepts are generated to complete the whole sketch graph. We achieve cross-concept references by argument passing (see inset above for illustration). In particular, we implement the cross-concept composition operator $R_S$ as an assignment matrix $\mathbf{R}_S$ of shape $(k_{qry} \cdot k_{arg}) \times (k_{qry} \cdot k_{arg})$ directly mapped from $\mathbf{q}_R$ through an MLP. Similar to the in-concept composition matrix, each row of the cross-concept matrix is a discrete distribution such that $\sum_j \mathbf{R}_S[i, j] = 1$, with the maximum entry signifying that the $(i \bmod k_{arg})$-th outward argument of the $\lfloor i/k_{arg} \rfloor$-th concept instance refers to the $(j \bmod k_{arg})$-th inward argument of the $\lfloor j/k_{arg} \rfloor$-th concept instance.

The complete cross-concept reference is therefore the product of three transport matrices:

$$\mathbf{R}_{cref}[t_i^1, t_j^1] = \mathbf{R}_{t_i^1}[{:}2k_{L^0}, k_{L^0}{:}] \times \mathbf{R}_S[i \cdot k_{arg}{:}(i{+}1) \cdot k_{arg}, j \cdot k_{arg}{:}(j{+}1) \cdot k_{arg}] \times \mathbf{R}_{t_j^1}[2k_{L^0}{:}, {:}k_{L^0}],$$

$$(2)$$

where $\mathbf{R}_{cref}[t_i^1, t_j^1]$ is a block assignment matrix of shape $2k_{L^0} \times k_{L^0}$. Intuitively, $\mathbf{R}_{cref}[t_i^1, t_j^1]$ specifies how constraints inside $t_i^1$ refers to primitives of $t_j^1$ throughout all possible paths crossing the arguments of two concepts.

Collectively, we denote the complete reference matrix for all pairs of generated $\mathbb{L}^0$ elements as $\mathbf{R}$ of shape $(2k_{qry} \cdot k_{L^0}) \times (k_{qry} \cdot k_{L^0})$, which includes in-concept and cross-concept references.

### 4.3 Concept instantiation by parameter generation

Instantiating a concept structure requires assigning parameters to the components where applicable. Therefore, as shown in Fig. 2, the parameter generation network takes a concept structure $\mathbf{T}^1$ and its implicit instance encoding $\mathbf{q}$ as input, and produces the specific parameters for each $\mathbb{L}^0$ typed instances inside the concept. In addition, as the parameters of generated instances are directly related to the input parameters of the raw sketch $S$, we find it improves convergence and accuracy by allowing the parameter network to attend to the input tokens.

We implement the parameter network as a transformer decoder in a similar way as [2]. The instance code $\mathbf{q}$ is first expanded to $k_{L^0}$ tokens by a small MLP, which are summed with $[\mathbf{t}_i^0]$ token-wise to obtain the query codes. The parameter network then transforms the query codes through interleaved layers of self-attention, cross-attention to the contextualized input sequence $[\mathbf{e}_{t^0}']$, and feed-forward, before finally mapped to explicit parameters in the form of probabilities over quantized bins, through a decoding layer $\mathrm{dec}_{param}(\cdot)$ that is inverse of $\mathrm{enc}_{param}(\cdot)$ in Sec. 4.1.

## 5 Induction objectives

Without any given labels of concepts, we use the following objectives to supervise the inductive network training: sketch reconstruction, concept quantization, and modularity enhancement.

### 5.1 Reconstruction loss

As discussed in Sec. 4, an input sketch is restructured into a set of sketch concepts which are expanded into a graph of primitives and constraints; the generated sketch graph $\widetilde{S}$ is compared with the input sketch $S$ for reconstruction supervision.

The comparison of generated and target graphs requires a one-to-one correspondence between elements of the two graphs, on which the graph differences can be measured. However, it is nontrivial to find such a matching, because not only are there variable numbers of elements in the two graphs, but also both *elements* and *references between elements* must be taken into account for matching. To this end, we build a cost matrix that measures the difference for each pair of generated and target elements, in terms of their attributes and references, and apply linear assignment matching on the cost matrix [11, 8] to establish the optimal correspondence between two graphs.

**Cost matrix construction** To compare each pair of generated element and target element, we measure their type differences, and further use type casting to interpret the generated element as the target type, so that their parameters can be compared. To account for reference differences between the two elements, we compare the reference arrows by the differences of their pointed primitives.

Specifically, given the target graph $S$ of $k_{tgt}$ elements and the generated graph $\widetilde{S}$ of $k_{qry} \cdot k_{L^0}$ elements, we build the cost matrix $\mathbf{C}$ of shape $k_{tgt} \times (k_{qry} \cdot k_{L^0})$ in two steps. First, for a pair of target element $p$ and generated element $q$, we compare their type and parameter differences by cross-entropy. We denote the cost matrix in this stage as $\mathbf{C}_{ury}$, as it accounts for the element-wise unary distances between two graphs. Second, to measure the binary distances of references, for each target constraint element $p$ and its $r$-th referenced primitive $p.r$, its distance from the generated references of element $q$ is computed as (also illustrated by inset figure):

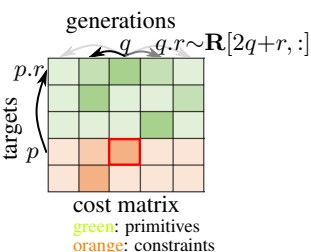

generations

cost matrix
green: primitives
orange: constraints

Binary cost between target constraint $p$ and generated element $q$.

$$\mathbf{C}_{bry}[p, q] = \sum_{r \in \{0,1\}} \sum_{j \in \widetilde{S}} \mathbf{R}[2q + r, j] \times \mathbf{C}_{ury}[p.r, j], \tag{3}$$

where $\mathbf{R}[2q+r, j]$ is the probability of $q$ taking $j$ as its $r$-th reference, as predicted by the composition operation (Sec. 4.2). Intuitively, the binary cost is a summation of unary costs weighted by predicted reference probabilities, where the unary costs measure how different a generated pointed primitive is from the target pointed primitive. The complete cost matrix is $\mathbf{C} = w_{ury}\mathbf{C}_{ury} + w_{bry}\mathbf{C}_{bry}$, with $w_{ury} = 50, w_{bry} = 1$; we give a larger weight to the unary costs because meaningful binary costs depend on reliable unary costs in the first place, as evident in Eq. (3).

**Matching and reconstruction loss** Given the cost matrix $\mathbf{C}$, we apply linear assignment to obtain a matching $\sigma : \widetilde{S} \to S \cup \{\phi\}$ between $\widetilde{S}$ and $S$. Note that the number of elements of these two graphs can be different, but we have chosen $k_{qry}, k_{L^0}$ such that the generated elements always cover the target elements. Therefore, $\sigma(q) = p$ assigns a matched generation $q \in M \subset \widetilde{S}$ to a target $p \in S$, but assigns the rest unmatched generations $M' = \widetilde{S} \backslash M$ to the empty target $\phi$, i.e. $\sigma(M') = \phi$. The loss terms for matched generations are simply the corresponding cost terms $\mathbf{C}[\sigma(q), q], q \in M$; for unmatched generations, we supervise its type to be the empty type $\phi$ and neglect its parameters or references. We denote the average loss of all generated terms as $\mathcal{L}_{recon}$.

Besides matching cost, we also use an additional reference loss to encourage the generated references to be sharp (i.e., $\mathbf{R}$ being closer to binary). This loss complements the binary costs mentioned above by making sure that even if the generated primitives are similar, a generated constraint only refers to one primitive sharply. We define the sharp reference loss as

$$\mathcal{L}_{sharp} = -\frac{1}{|S_c|} \sum_{p \in S_c, r} \log \mathbf{R}[2\sigma^{-1}(p) + r, \sigma^{-1}(p.r)], \tag{4}$$

where $p$ iterates over the target constraints $S_c$, $\sigma^{-1}(\cdot)$ is the inverse mapping from target element to generation, and we skip a term if $p.r$ does not exist for constraints with one reference.

## 5.2 Concept quantization loss

Following [21, 16], we optimize the concept code quantization against library $\mathbf{L}^1$ with:

$$\mathcal{L}_{vq} = \frac{1}{k_{qry}} \sum_{i \in [k_{qry}]} ||\mathrm{sg}(\mathbf{q}_i) - \mathbf{q}'_i|| + \beta ||\mathbf{q}_i - \mathrm{sg}(\mathbf{q}'_i)||, \tag{5}$$

where $\mathrm{sg}(\cdot)$ is the stop gradient operation. For training stability, we follow [16] and replace the first term with EMA updates of $\mathbf{q}' \in \mathbf{L}^1$. Furthermore, we improve spare code usage by reviving unused code in $\mathbf{L}^1$ periodically [16] (please refer to supplementary for details).

## 5.3 Modularity enhancement loss

We look for modular $\mathbf{L}^1$ concepts that have rich and meaningful encapsulated structures, rather than arbitrary groups of $\mathbb{L}^0$ elements that rely on cross-group references to recover the graph structures. This modularity can be enhanced by limiting the use of arguments for sketch concepts. Instead of allocating very few arguments as a hard constraint, we introduce a soft bias loss to encourage the restrictive use of arguments, which may still cover cases when more arguments are needed for accurate reconstruction. To be specific, we penalize the accumulated probability of elements pointing to outward arguments:

$$\mathcal{L}_{bias} = \frac{1}{|S_c|} \sum_{p \in S_c, r} \sum_{i \in [k_{arg}]} \mathbf{R}_{\mathbf{T}^1 \ni \sigma^{-1}(p)}[2\sigma^{-1}(p) + r, i + k_{L^0}], \tag{6}$$

where again $p$ iterates over the target constraints $S_c$, $\sigma^{-1}(\cdot)$ is the inverse mapping from target element to generation, and $i + k_{L^0}$ slices the reference probabilities to outward arguments.

## 5.4 Total loss

The training objective sums up losses of reconstruction, concept quantization and modularity bias:

$$\mathcal{L}_{total} = w_{recon}\mathcal{L}_{recon} + w_{sharp}\mathcal{L}_{sharp} + w_{vq}\mathcal{L}_{vq} + w_{bias}\mathcal{L}_{bias}, \tag{7}$$

where we empirically use weights $w_{recon} = 1, w_{sharp} = 20, w_{vq} = 1, w_{bias} = 25$ throughout all experiments unless otherwise specified in the ablation studies.

# 6 Results

**Dataset and implementation**  Following previous works [6, 13, 18], we adopt the SketchGraphs dataset [17] which contains millions of real-world CAD sketches for training and evaluation. We filter the data by removing trivially simple sketches and duplicates, and limit the sketch complexity such that the number of primitives and constraints is within $[20, 50]$. As a result, we obtain around 1 million sketches and randomly split them into 950k for training and 50k for testing. We defer network details to the supplementary and open-source code and data to facilitate future research[3].

**Evaluation metrics**  We evaluate the generated sketches in terms of reconstruction accuracy and sketch concept modularity, which are the two major objectives of our task. We measure reconstruction accuracy by the F-scores of generated primitives and constraints, where F-score is simply the harmonic mean of precision and recall. A generated primitive is considered a correct match with ground-truth if its type and parameters are correct, where for the scalar parameters we allow a threshold of $10\%$ of quantization levels. A constraint is correct if and only if its type, parameter and references match ground-truth, i.e., the generated $q$ is correct w.r.t target $p$ iff $q$ has the same type and parameters with $p$ and the primitives $q.r$ and $p.r$ are correctly matched. Modularity is measured by the percentage of constraints with references entirely within the encapsulating concepts, among all correct constraints.

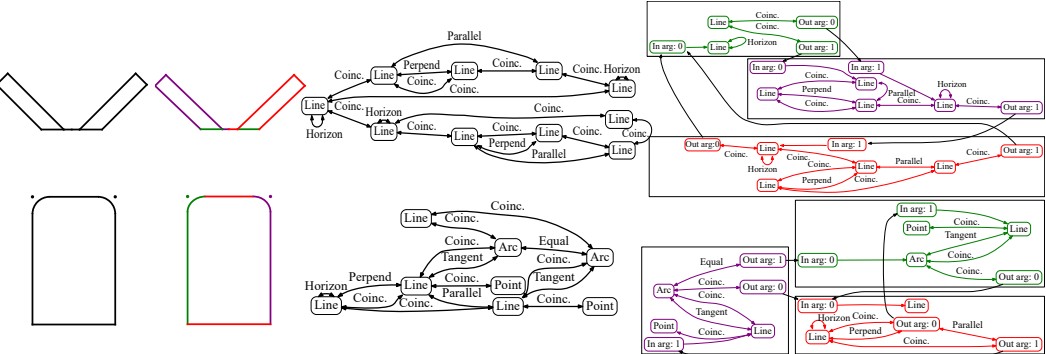

Figure 3: **Design intent parsing.** Left: input raw sketches and sketches restructured with concepts. Right: raw constraint graphs and modular constraint graphs. Primitives and constraints in the restructured sketches and graphs are colored according to concepts.

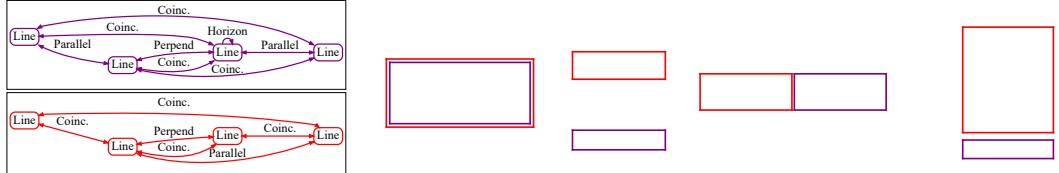

Figure 4: **Instances of concepts.** Left: two learned rectangle concepts with subtly different structures. Right: four sketches containing instances of these two concepts.

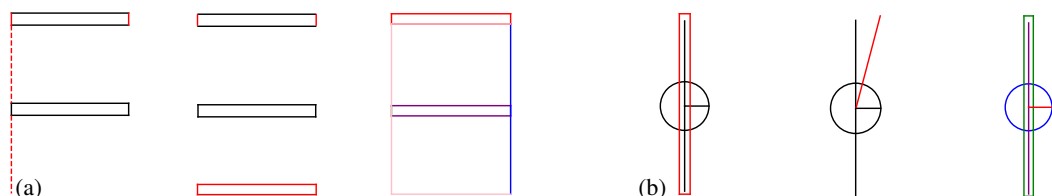

Figure 5: **Auto completion.** Each example shows the input partial sketch (black) and groundtruth completion (red), result of the autoregressive baseline, and our result (colored by concepts).

---

[3]URL to code and data: `https://github.com/yyuezhi/SketchConcept`

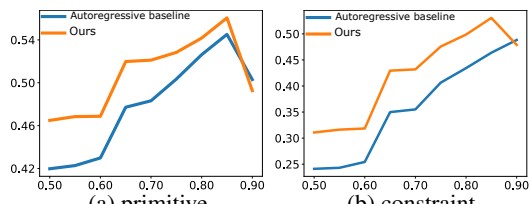

Figure 6: **Auto completion comparison.** Plotted are F-scores at different ratios of partial input.

Table 1: **Loss ablation.** F-scores are reported for primitives and constraints.

| Config | Primitive | Constraint | Modular(%) |
|---|---|---|---|
| $-\mathbf{C}_{bry}$ | 0.993 | 0.290 | 34.2 |
| $-\mathcal{L}_{sharp}$ | 0.983 | 0.104 | 100 |
| $-\mathcal{L}_{bias}$ | 0.992 | 0.743 | 13.3 |
| Ours | 0.994 | 0.766 | 50.8 |

## 6.1 Design intent interpretation

By training our model on the raw sketches with self-supervised induction losses, we obtain a result library of sketch concepts and a model for design intent parsing that interprets a given sketch into modular concepts and their combination. Indeed, we find the automatically discovered concepts capture natural design intents and modular structures. For example, through the restructured sketches and constraint graphs in Figs. 1 and 3, we find that our network decomposes sketches into modular structures like rectangles, line-arcs and parallel lines that align symmetrically, even though no such prior knowledge is applied during training except for concept modularity. Fig. 4 shows that a given concept can be used repetitively in different sketches, and structures with subtle differences in constraint relations can be detected and distinguished into different concepts of the library. Note that these subtle structural differences are subsumed in the input sketch graph, which makes them more difficult to detect. We refer to the supplementary for more examples of design intent parsing and instantiation of learned concepts, as well as quantitative analysis of the learned library.

## 6.2 Auto completion

Auto-completion is a critical feature of CAD modeling software for assisting designers. Given a partial sketch of primitives and their constraints, auto-completion aims at complementing them with the rest primitives and constraints to form regular and well-structured designs. Therefore, our concept detection and generation approach would naturally enhance the auto-completion task with better regularity. For training and evaluation, following previous work [18], we synthesize the partial input by removing a suffix of random length (up to $50\%$) from the sketch sequence, along with constraints that refer to the removed primitives, and make the model learn to generate the full sketches.

State-of-the-art methods [6, 13, 18, 24] formulate auto-completion through a combination of primitive and constraint generation models, both of which operate in an autoregressive fashion, with the constraint model conditioned on and referring (by pointers [23]) to the generated primitives. Since these works use diverse sketch encodings and have no publicly released code at submission time, for fair comparison, we implement the autoregressive baseline with our sketch encoding (Sec. 4.1).

Fig. 6 compares our method with autoregressive baseline under various primitive mask ratios: our method has superior primitive and constraint accuracy than the autoregressive baseline at almost all mask ratios. This difference confirms that since our method completes sketches concept-by-concept instead of primitive-by-primitive, more meaningful structures are likely to be generated. Our model also gains advantage by taking primitives and constraints together as input and generating primitives and constraints simultaneously, while in comparison the autoregressive baseline separates generation in two steps (primitives followed by constraints). Indeed, in practice CAD designers rarely finish all primitives first before supplementing the constraints, but rather apply constraints on partial primitives immediately whenever they form a design intent. Fig. 5 shows how our approach interprets the partial inputs and completes with modular concepts (see supplementary for more examples); in comparison, the autoregressive baseline does not provide such interpretable or regular completions.

## 6.3 Ablation study

To evaluate the impact of different loss terms of the induction objective (Sec. 5), we train several models in the absence of these losses respectively on the auto-encoding task. The results are shown in Table. 1. We see that removing the binary costs $\mathbf{C}_{bry}$ from reconstruction loss results in significant drop of constraint reconstruction, showing its necessity for constraint reference learning. Removing sharp reference loss $\mathcal{L}_{sharp}$ similarly fails constraint reference learning, although modularity

enhancement bias loss makes all constraint references inside concepts. Removing the modularity enhancement bias loss $\mathcal{L}_{bias}$ only results in a slight drop in reconstruction quality but a significant drop in modularity, since without it cross-concept reference through arguments is more likely and therefore modularity suffers. We provide more ablation tests on hyper parameters like the numbers of concept queries $k_{qry}$ and arguments $k_{arg}$ in the supplementary.

## 7    Conclusion

CAD sketch concepts are meta-structures containing primitives and constraints that define modular sub-graphs and capture design intents. By formulating the sketch concepts as program libraries of a DSL, we present an end-to-end approach for discovering CAD sketch concepts through library induction learning. Key to our approach are the implicit-explicit representation of concepts and the separated structure generation and parameter instantiation for concept generation, which together enable the end-to-end training under self-supervised induction objectives. By training on large-scale sketch dataset, our approach enables the discovery of repetitive and modular concepts from raw sketches, and more structured and interpretable auto-completion than baseline autoregressive models.

**Limitations and future work**  Design intents can be hierarchical [10], meaning that higher order meta-structures can be built out of lower order ones. In this sense, our framework only addresses the first order library induction, and should be extended for higher order library learning; toward this goal, we believe a progressive approach like [5] can be used with our framework as the one-step induction. In addition, similar strategies of end-to-end induction learning can be applied to constraint graphs involving 3D CAD operations or even more general programs in other domains, as long as they have similar declarative and parametric structures as sketch graphs.

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
