# A  Supplementary for "Discovering Design Concepts for CAD Sketches"

## A.1  The complete list of $\mathbb{L}^0$ types

We provide the complete list of $\mathbb{L}^0$ types in List 2. These types are constructed based on the given data types from the SketchGraphs dataset [17]. Note that in the current implementation we do not distinguish sub-primitive references that point to different parts of a primitive, but rely on the predicted geometric closeness of primitive parts to tell them in the post-process, as we find the geometric predictions are generally quite accurate for this purpose. On the other hand, we note that the extension of references into primitive parts can be trivially achieved by turning primitives into functions and augmenting them with arguments (similar to how we model constraints), such that each argument corresponds to a primitive part; the constraint references can then pinpoint to primitive parts through argument passing (Sec. 4.2).

---

**List 2:** The complete list of $\mathbb{L}^0$ types considered in this work.

```
// Basic data types
```
Construction, Length, Angle, Coord, Ref
```
// 𝕃⁰ primitive types
```
Line $\rightarrow b_{dash}$: Construction, $c_{start\_x}, c_{start\_y}, c_{end\_x}, c_{end\_y}$ : Coord
Circle $\rightarrow b_{dash}$: Construction, $c_{center\_x}, c_{center\_y}$ : Coord, $l_{radius}$ : Length
Point $\rightarrow b_{dash}$: Construction, $c_x, c_y$ : Coord
Arc $\rightarrow b_{dash}$: Construction, $c_{center\_x}, c_{center\_y}$ : Coord, $l_{radius}$ : Length , $a_{start}, a_{end}$ : Angle
```
// 𝕃⁰ constraint types
```
Coincident $\rightarrow \lambda(r_1, r_2 : \text{Ref}).\{\}$
Distance $\rightarrow \lambda(r_1, r_2 : \text{Ref}).\{l_{dist} : \text{Length}\}$
Horizontal $\rightarrow \lambda(r_1 : \text{Ref}).\{\}$
Parallel $\rightarrow \lambda(r_1, r_2 : \text{Ref}).\{\}$
Vertical $\rightarrow \lambda(r_1 : \text{Ref}).\{\}$
Tangent $\rightarrow \lambda(r_1, r_2 : \text{Ref}).\{\}$
Length $\rightarrow \lambda(r_1 : \text{Ref}).\{l_{dist} : \text{Length}\}$
Perpendicular $\rightarrow \lambda(r_1, r_2 : \text{Ref}).\{\}$
Equal $\rightarrow \lambda(r_1, r_2 : \text{Ref}).\{\}$
Diameter $\rightarrow \lambda(r_1 : \text{Ref}).\{l_{dist} : \text{Length}\}$
Radius $\rightarrow \lambda(r_1 : \text{Ref}).\{l_{dist} : \text{Length}\}$
Angle $\rightarrow \lambda(r_1, r_2 : \text{Ref}).\{a_{ang} : \text{Angle}\}$
Concentric $\rightarrow \lambda(r_1, r_2 : \text{Ref}).\{\}$
Normal $\rightarrow \lambda(r_1, r_2 : \text{Ref}).\{\}$

---

## A.2  Implementation details

**Sketch encoding format**  In Sec. 4 we described how sketches are encoded to allow network learning; here we present more implementation details.

We encode the input sketch $S$ as a series of primitive tokens followed by a series of constraint tokens, with these tokens supplemented by learned positional encoding according to their indices in this sequence (Sec. 4.1). We additionally insert learnable START, END and NEW tokens at the front of the sequence, the end of the sequence, as well as between every encoded primitive/constraint respectively, to produce the complete sequence.

Each primitive is represented by two consecutive tokens: a $\mathbb{L}^0$ type token and a parameter token. The $\mathbb{L}^0$ type of primitive is encoded by a 256-dim embedding, obtained by an embedding layer denoted as $\text{enc}_{type}$. The parameters of a primitive are encoded in the parameter token; compared with using different numbers of tokens for different primitive types that previous autoregressive baselines do [6, 13, 18], our one-token parameter encoding allows straightforward matching with a target primitive even if the predicted primitive type does not match the target, which simplifies training. In particular, we use a schema shown in Fig. 7 to encode the parameter values, where each basic data type is represented by a 14-dim code that is obtained by embedding the quantized parameter value, and all slots of a specific primitive type are used while the rest slots are set zero. To represent that the

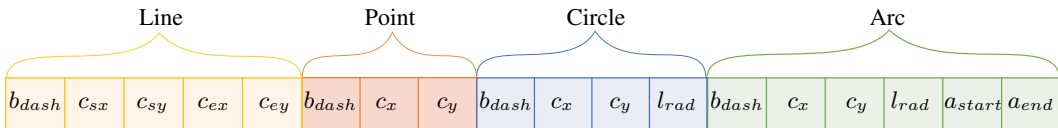

Figure 7: **Parameter code schema**. A parameter code contains 18 tokens, each of 14 dims, that are concatenated and zero-padded to 256 dims. For a particular primitive type, only tokens corresponding to the specified type are used in the parameter code, the rest tokens are reset to zero. We did not allocate slots for constraint parameters in the current implementation; in comparison, previous works [13, 18] simply omit constraints with parameters.

resultant parameter token belongs to a specific primitive type, we augment parameter token with the type token by summing the two vectors to produce the final parameter token (Sec. 4.1).

Each constraint is represented by a type token and several reference tokens. The constraint type token is obtained through the same embedding layer $enc_{type}$ as primitives. To encode a reference, we use a 256-dim embedding to encode the primitive index in this expanded sequence, obtained through the embedding layer $enc_{ref}$. We omit constraint parameters in the current implementation, and defer their inference to post-processing according to the positions of predicted primitives; in comparison, most previous works [13, 18] have simply skipped constraint types with parameters.

**Sketch parameter decoding** $dec_{param}()$ has a mirrored structure of $enc_{param}()$. It takes a latent parameter code as input and decodes it into a 256-dim code (Fig. 7), which contains several segments corresponding to different primitive types. When doing type casting (Sec. 5), the segment corresponding to the target type is taken for parameter decoding. Each primitive property is represented by a 14-dim embedding code, from which a quantized property value is recovered by an inverse-embedding layer; during this inverse-embedding process, the logits are processed by argmax to query the quantized value. Following previous works [6, 13, 18], we always work with quantized attribute values as categorical variables during network training and inference.

**Normalization, augmentation and quantization** We normalize all sketches inside a $2 \times 2$ square centered at the origin, and remove duplicated sketches by rasterizing into $128 \times 128$ binary valued images and removing those with the same images. We apply random shrinking augmentation with scaling factors of $0.5 \sim 0.8$. The continuous basic data types (List 2) are uniformly quantized; in particular, we assign 30 bins for *angle*, 20 bins for *length* and 80 bins for *coordinate*.

**Network and training details** The detection network is a transformer encoder-decoder network, with the encoder/decoder having 12 layers, 8 attention heads and latent dimension of 256.

The structure generation network takes a library code $\mathbf{q}' \in \mathbf{L}^1$ and generates the $\mathbb{L}^0$ type elements $[t_i^0]$ within and a matrix representing the composition $R_{\mathbf{T}^1}$ of $[t_i^0]$ and arguments. Specifically, the 256-dim library code $\mathbf{q}'$ first passes through an MLP[4] of 3 layers to expand to $k_{L^0} \times 256$ dims, i.e. $k_{L^0}$ codes representing $[t_i^0]$ elements. Then each code passes through another MLP of 3 layers (i.e. $dec_{type}$) to output the discrete probabilities of $\mathbb{L}^0$ types that $t_i^0$ assumes. To generate the composition matrix $\mathbf{R}_{\mathbf{T}^1}$, we use another MLP of 5 layers to expand the library code to a $(2k_{L^0}+k_{arg}) \times (k_{L^0}+k_{arg})$ matrix and apply softmax on each row, as detailed in Sec. 4.2.

The parameter network generates parameters to instantiate concepts. It first expands each of the $k_{qry}$ concept instance codes $[\mathbf{q}_i]$ into $k_{L^0}$ parameter latent codes, which are further added with the corresponding parameter type embeddings obtained from structure generation network and fed into a transformer decoder to generate explicit parameters. The transformer decoder here has the same hyper parameters as the concept detection decoder (i.e. 12 layers, 8 attention heads, and 256 latent dimension). The decoder transforms each group of $k_{L^0}$ latent parameter codes by cross-attending to contextualized input sequences $[\mathbf{e}'_{t_i^0}]$, and finally maps them to parameter tokens as described in Fig. 7 through $dec_{param}$, which are further decoded into probabilities over quantized basic data types by corresponding inverse embedding layers.

We implement all modules in Pytorch, and use the Adam optimizer with a learning rate of $10^{-4}$ to train the network for 160 epochs on 4 V100 GPUs, which takes 2 days to complete.

---

[4]Unless otherwise specified, all MLPs used in this paper have uniform hidden dimensions as the input dimension and ReLU activation after each hidden linear layer.

**Library size, EMA code update and dead code revival** In our experiments, we use a library of 1000 candidate concepts for $\mathbf{L}^1$. We follow [16] to replace the first term of concept quantization loss (i.e. $\|\text{sg}(\mathbf{q}_i) - \mathbf{q}'_i\|$ ) with exponential moving average (EMA) updates of $\mathbf{q}' \in \mathbf{L}^1$. Specifically, for each code $\mathbf{q}'_i$, we define two accumulated variables $n_i \geq 0$ and $\mathbf{m}_i \in \mathbf{R}^d$, which are initialized as 1 and a random unit vector, respectively. They are later updated in each gradient descent iteration following the rules:

$$n_i := \gamma n_i + (1 - \gamma)N_i \tag{8}$$

$$\mathbf{m}_i := \gamma \mathbf{m}_i + (1 - \gamma)\sum_j \mathbf{q}_{i,j} \tag{9}$$

$$\mathbf{q}'_i := \frac{\mathbf{m}_i}{n_i} \tag{10}$$

where $\{\mathbf{q}_{i,j}\}$ are $N_i$ detection queries that select $\mathbf{q}'_i$ as the closet concept prototype in this iteration. We set the decay rate $\gamma = 0.99$ and the commitment cost coefficient $\beta = 1$ in all our experiments.

In addition, we find that the concept quantization process may suffer from codebook collapse where all $[\mathbf{q}_i]$ select to few codes of $\mathbf{L}^1$, which impairs the capability of the model. To solve this problem, in the training process we use dead-code revival [16] to periodically (every 100 mini-batches) find an unused code in $\mathbf{L}^1$ and replace it with the $\mathbf{q}$ who has farthest distance to its closest code $\mathbf{q}'$.

### A.3 Autoregressive baseline implementation detail

Following [24, 13, 18], the autoregressive baseline contains two modules, the primitive model that generates primitives sequentially and the constraint model that takes primitives as input and generates constraints sequentially. The primitive model is an autoregressive transformer decoder of 12 layers, 8 attention heads and latent dimension 256. The constraint model is a transformer encoder-decoder, where the encoder contextualizes input primitives, and the decoder is an autoregressive model generating constraints. Constraint reference to primitives is implemented by computing dot product correlation between the generated reference token and contextualized primitive tokens produced by the encoder, following the Pointer Network design [23]. The constraint model encoder/decoder have the same hyper-parameters as the primitive model.

### A.4 More results

We present more results on design intent interpretation and auto completion. In Fig. 8, we show more results of how raw sketches are parsed with learned concepts, where primitives are colored according to their encapsulating concepts, and constraint graphs are visualized to show the modular concepts. In Fig. 9 we show more such design intent interpretation results without constraint graphs. Fig. 10 presents more auto-completion results, where again we compare with baseline autoregressive approach and demonstrate better interpretability and more regular completions.

### A.5 Concept library analysis

Fig. 11(a) shows the frequency of how often our learned library concepts are used in the test dataset. The distribution shows a long-tail pattern, which is expected because the diversity of sketches demands a wealth of modular sketch concepts that individually may not be used extensively. We provide more concrete concepts and corresponding sketches containing these concepts in Fig. 12. These concepts are arranged according to appearance frequency (from high to low) as marked with red points in Fig. 11(a). The most frequently used concepts are rectangles with different constraint variants due to their high abundance within regular sketches. Besides, concepts with simple structures, e.g. few lines connected together by coincidence, are generally more frequently used than those with complex structures, as the simple structures are more flexible and can fit in diverse sketches.

Fig. 11(b) shows the complexity of learned library concepts in terms of how many $\mathbb{L}^0$ instances are contained in a concept. We can see that there are a small number of degenerate concepts with empty $\mathbb{L}^0$ instances and trivial concepts with only one $\mathbb{L}^0$ instances. The empty concepts serve as placeholder for filling up the gaps between small sketches and the maximal graph of $k_{qry}$ concepts. The trivial concepts exist because we always convert a raw sketch into a set of $\mathbb{L}^1$ concepts, and for those $\mathbb{L}^0$ elements of the raw sketch that do not fit into any modular concept, they will be encapsulated with such trivial $\mathbb{L}^1$ concepts for the sake of complete reconstruction.

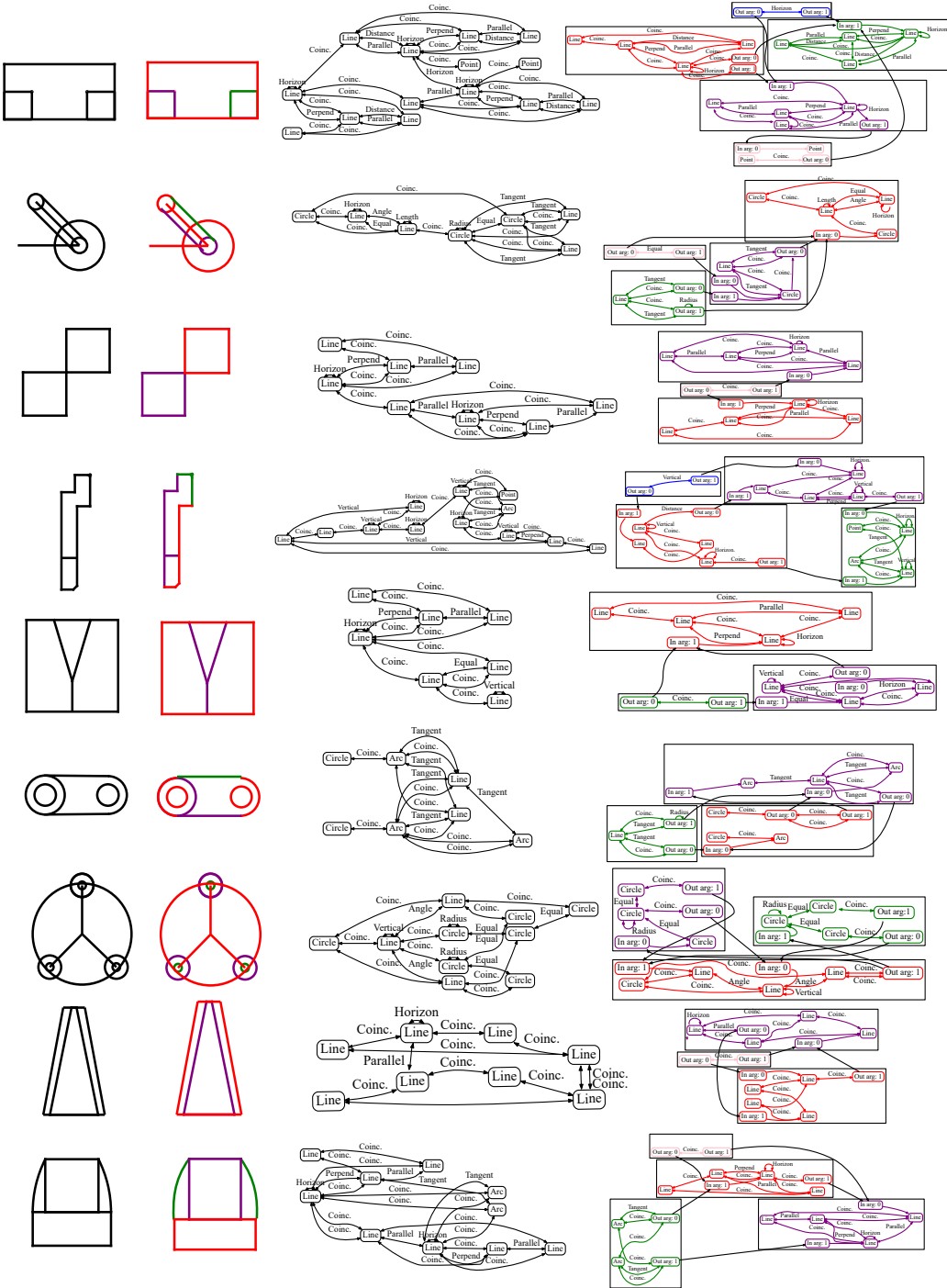

Figure 8: **Design intent parsing**. Each example shows the input raw sketch and corresponding constraint graph (in black), as well as our interpreted sketch made of modular concepts and corresponding modular constraint graph, where primitives and constraints are colored according to their encapsulating concepts.

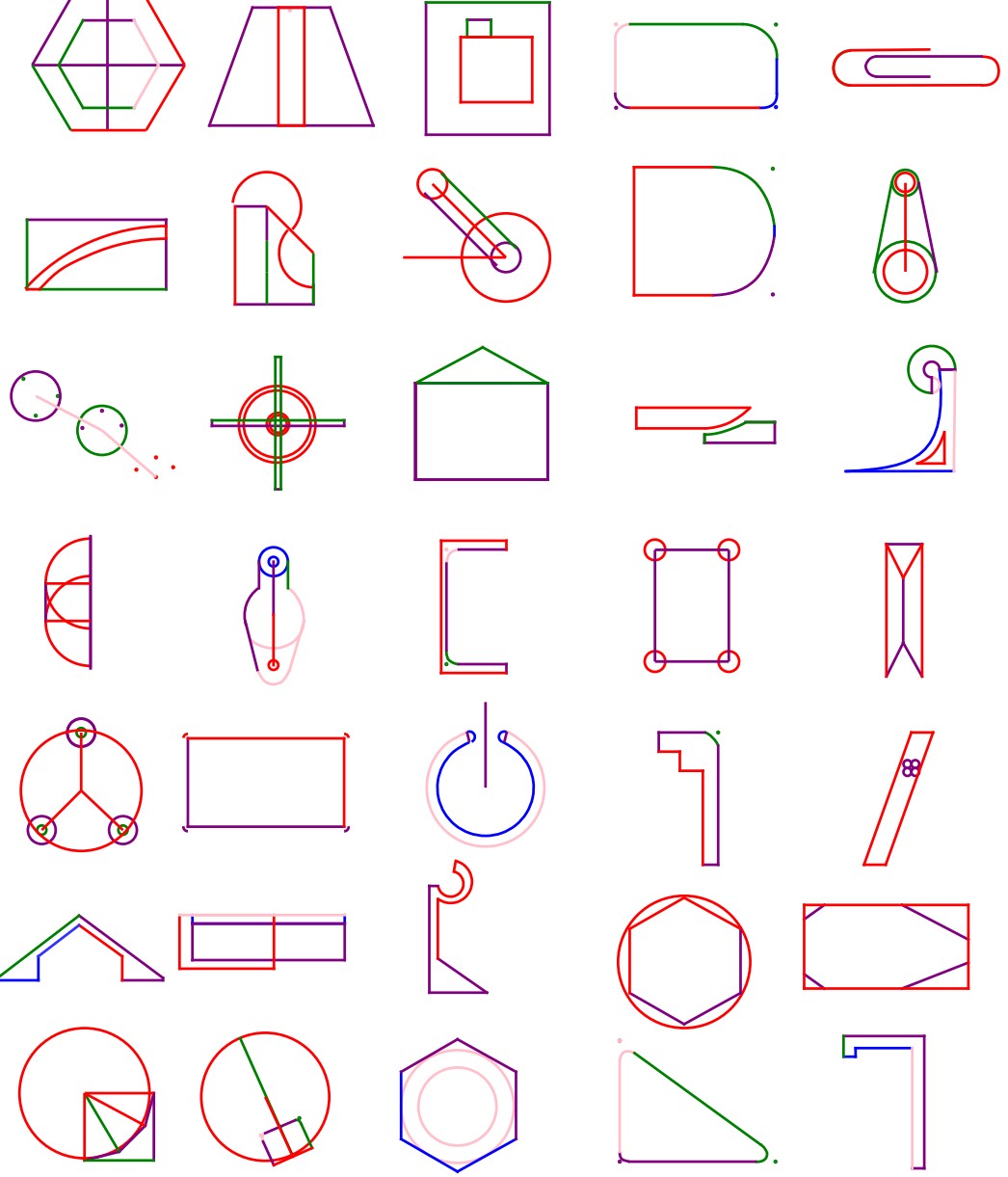

Figure 9: **Design intent parsing** without showing the constraint graphs. In each sketch example, $\mathbb{L}^0$ primitives of the same color belong to the same sketch concept.

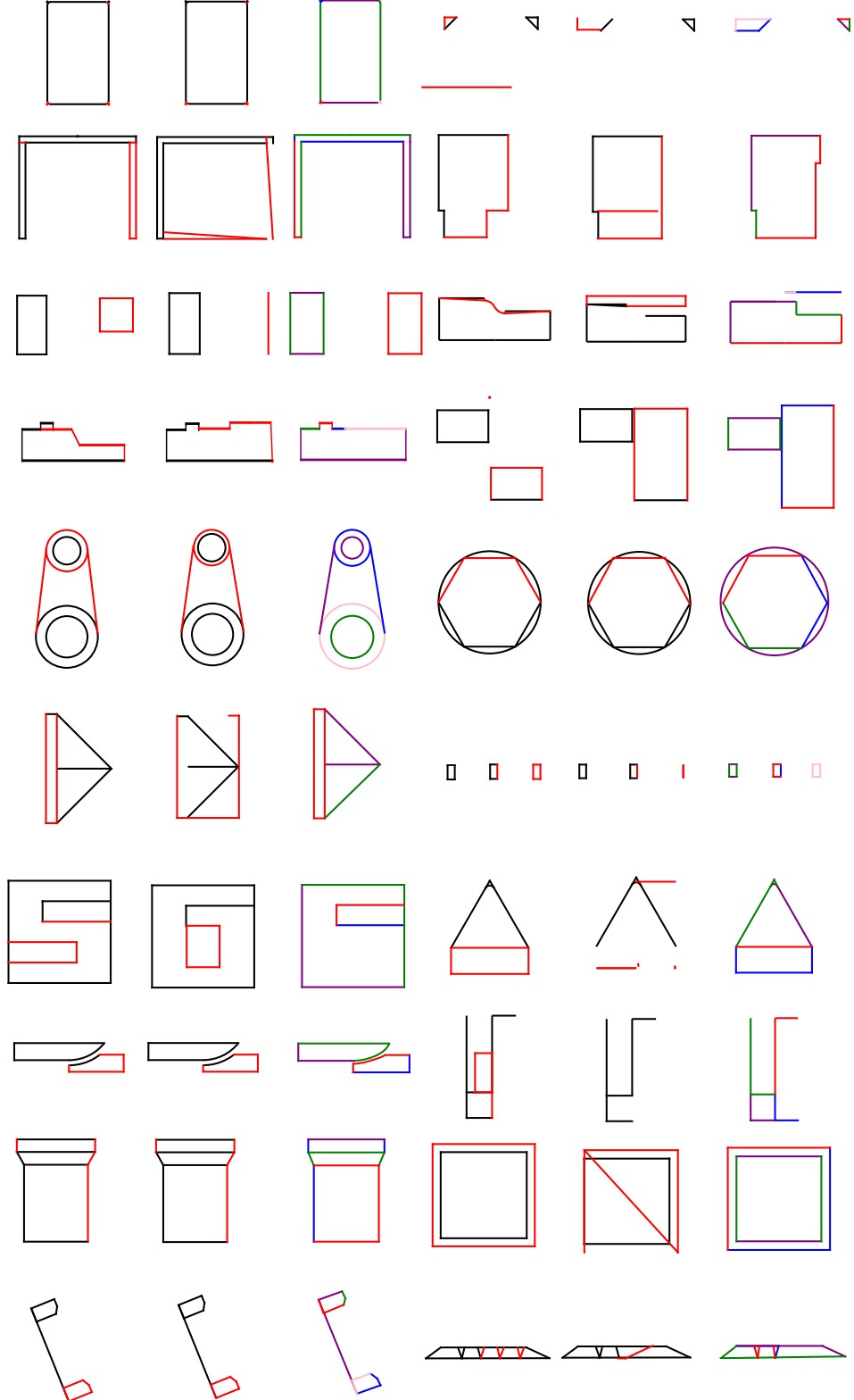

Figure 10: **More results of auto-completion**. Each example shows the input partial sketch (black) and groundtruth completion (red), result of the autoregressive baseline, and our result (colored by concepts). Our completion results show better interpretability and regularity.

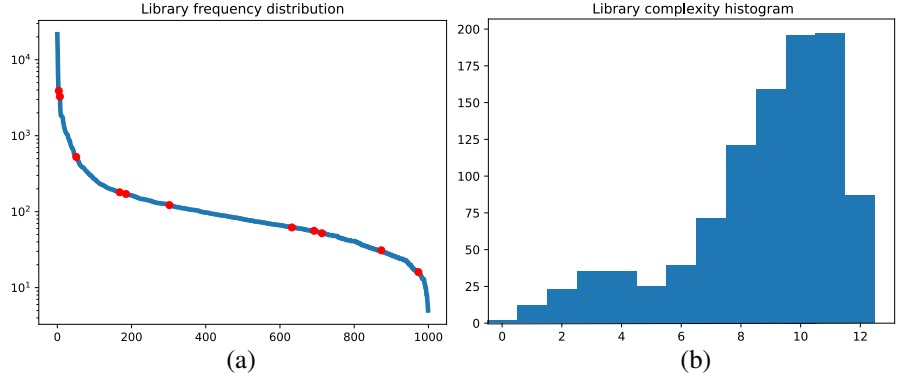

(a)

(b)

Figure 11: **Statistics about the learned library concepts**. **(a)** Library frequency distribution. The horizontal axis shows the 1000 $\mathbb{L}^1$ library concepts learned and sorted according to their frequencies in the test dataset. The vertical axis shows the frequency value in log-scale. The majority of learned concepts have stable but not very high occurrence frequencies, meaning they follow a long-tail distribution as expected. Concepts denoted by red points on the curve are visualized in Fig. 12. **(b)** Library complexity histogram. The horizontal axis is the number of $\mathbb{L}^0$ instances contained in a concept, and the vertical axis is the number of concepts of a specific complexity. We can see that degenerate (i.e. empty) or trivial (i.e. size 1) concepts are rare among the whole learned library.

## A.6 Parameter refinement with constraint solver

The errors in generated primitive parameters (e.g. due to quantization of basic data types) can be mitigated by applying constraints with a constraint solver provided by OnShape [15]. In Fig. 13, we show examples of sketches before and after refining primitive parameters with constraint solver.

## A.7 More ablation results

| $k_{qry}$ | Primitive | Constraint | Modular(%) |
|---|---|---|---|
| 5 | 0.994 | 0.766 | 50.8 |
| 6 | 0.994 | 0.808 | 36.0 |
| 8 | 0.991 | 0.845 | 32.6 |
| 10 | 0.998 | 0.894 | 15.9 |
| 12 | 0.999 | 0.918 | 14.9 |

Table 2: **Query number $k_{qry}$ ablation.** F-scores are reported for primitives and constraints.

| $k_{arg}$ | Primitive | Constraint | Modular(%) |
|---|---|---|---|
| 1 | 0.993 | 0.666 | 52.6 |
| 2 | 0.993 | 0.766 | 50.8 |
| 3 | 0.990 | 0.7577 | 18.5 |
| 4 | 0.994 | 0.776 | 7.1 |

Table 3: **Argument number $k_{arg}$ ablation.** F-scores are reported for primitives and constraints.

| $\mathbb{L}^1$ size | Primitive | Constraint | Modular(%) |
|---|---|---|---|
| 50 | 0.970 | 0.581 | 48.1 |
| 100 | 0.980 | 0.600 | 48.8 |
| 500 | 0.989 | 0.735 | 49.2 |
| 1000 | 0.994 | 0.766 | 50.8 |
| 2000 | 0.995 | 0.779 | 50.6 |

Table 4: $\mathbb{L}^1$ **library size ablation.** F-scores are reported for primitives and constraints.

To evaluate the impact of hyper parameters such as the number of concept queries $k_{qry}$, arguments $k_{arg}$ and library size $\mathbb{L}^1$, we train our model under different $k_{qry}$, $k_{arg}$ and $\mathbb{L}^1$ sizes on the auto-encoding design intent interpretation task.

When changing $k_{qry}$, we also adjust $k_{L^0}$ the number of $\mathbb{L}^0$ elements each concept contains, so that the total number of generated $\mathbb{L}^0$ elements, i.e. $k_{qry} \times k_{L^0}$, is unchanged. The evaluation results on adjusting $k_{qry}$, $k_{arg}$, $\mathbb{L}^1$ are shown in Table 2, Table 3 and Table 4 respectively.

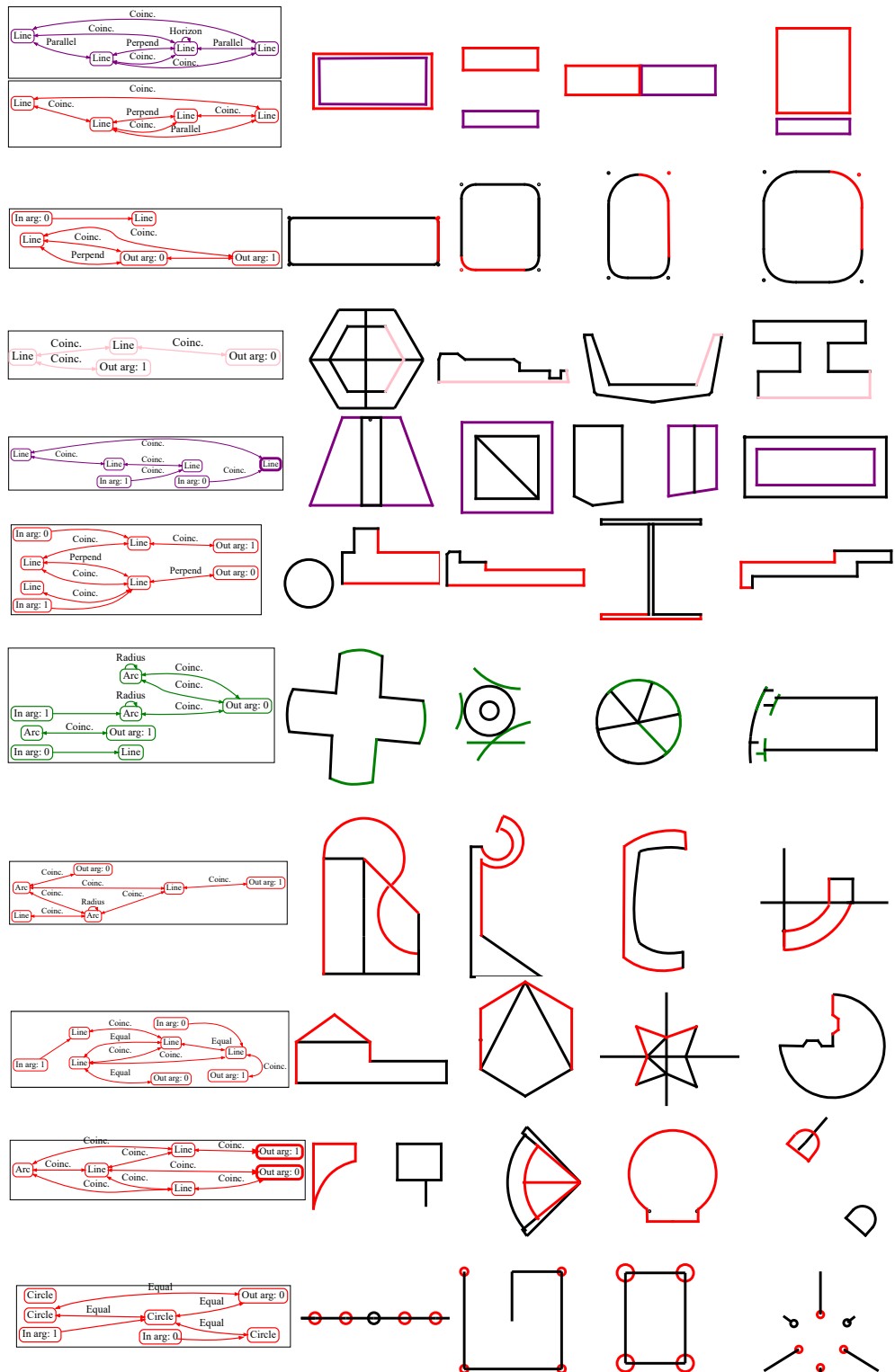

Figure 12: **Examples of learned library concepts and their corresponding sketch instances**. The concepts are sorted by their occurrence frequency (high to low) in the test set, and correspond to the red dots marked on the distribution curve of Fig. 11.

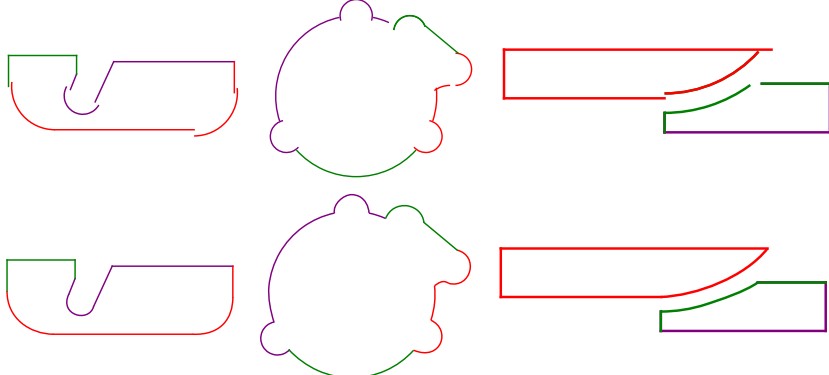

Figure 13: **Parameter refinement with constraint solver**. Generated sketches have parameter inaccuracies (upper row); constraint solver refines the sketches by applying generated constraints and fitting the primitives together properly (lower row).

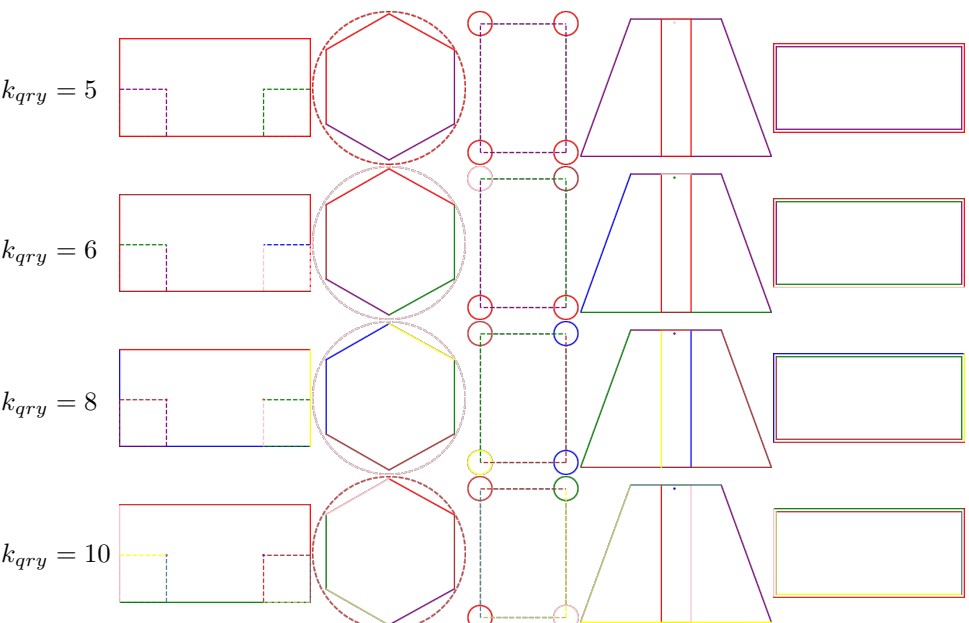

Figure 14: **Design intent interpretation trained with different concept query numbers** $k_{qry}$. Larger $k_{qry}$ leads to less modular concepts.

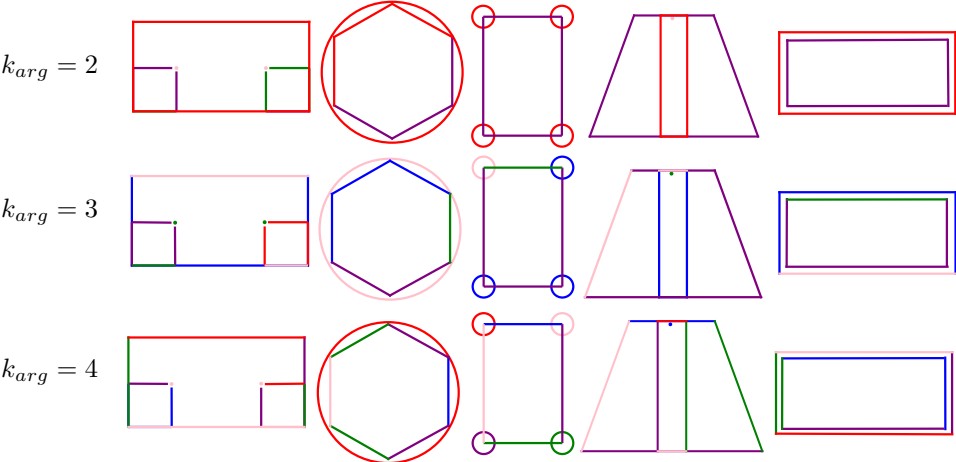

$k_{arg} = 2$

$k_{arg} = 3$

$k_{arg} = 4$

Figure 15: **Design intent interpretation trained with different argument numbers** $k_{arg}$. Larger $k_{arg}$ leads to less modular concepts.

From Table 2 we can see that adding more concept queries makes the model more expressive and flexible, demonstrated as the increasing constraint F-scores; on the other hand, this comes with a cost of hurting modularity, as a sketch can be decomposed into more granular components.

From Table 3 we see that adding more arguments $k_{arg}$ than default 2 does not result in a significant improvement in constraint F-score, but leads to a significant decrease in modularity, suggesting the current default argument number is sufficient. On the other hand, decreasing the number of arguments leads to a significant drop in constraint F-score but does not achieve obviously higher modularity.

We visualize examples of design intent interpretation for different $k_{qry}$ and $k_{arg}$ in Fig. 14 and Fig. 15 respectively, to give an intuitive sense of the above numeric results especially on modularity.

From Table 4 we can see that adding more $\mathbb{L}^1$ library makes the model more expressive and flexible, showing increasing constraint F-scores. However, such improvement becomes marginal when more libraries are introduced, as the newly introduced libraries are mainly used to capture structures that rarely appear and have little impact on the overall results (intuitively, they mainly continue the fall-off trend of far-right tail regions of the frequency distribution shown in Fig. 11(a)). On the other hand, the modularity maintains at roughly the same level throughout the changes over library size.

## A.8 Image-conditioned generation

We extend our model to image-conditioned generation, where we are interested in accurately recovering a parametric sketch from an image of hand-drawn sketch. For comparison, we also extend the auto-regressive baseline to this image-conditioned generation.

Following [13, 18], we use a ViT style encoder to condition the generation on images. Specifically, the input sketch image of size $128 \times 128$ is partitioned into non-overlapping square patches of size $16 \times 16$. The image patches are flattened and pass through an MLP of 3 layers to produce a sequence of 64 image tokens (each of dimension 256), and then feed into a transformer encoder to produce contextualized image embeddings that the detection decoder cross-attends to. For autoregressive baseline, we similarly augment the primitive model with such an image encoder and use cross-attention to image tokens in the autoregressive primitive decoder. The ViT style image encoder used here has the same hyper parameters as the other transformer modules discussed above (i.e. 12 layers, 8 attention heads, 256 latent dimension).

We train our model with learning rate of $3 \times 10^{-4}$ for 200 epochs and the autoregressive baseline with the same learning rate for 400 epoch to convergence. We used the xkcd packages in mathplotlib to simulate sketches of hand-drawn style.

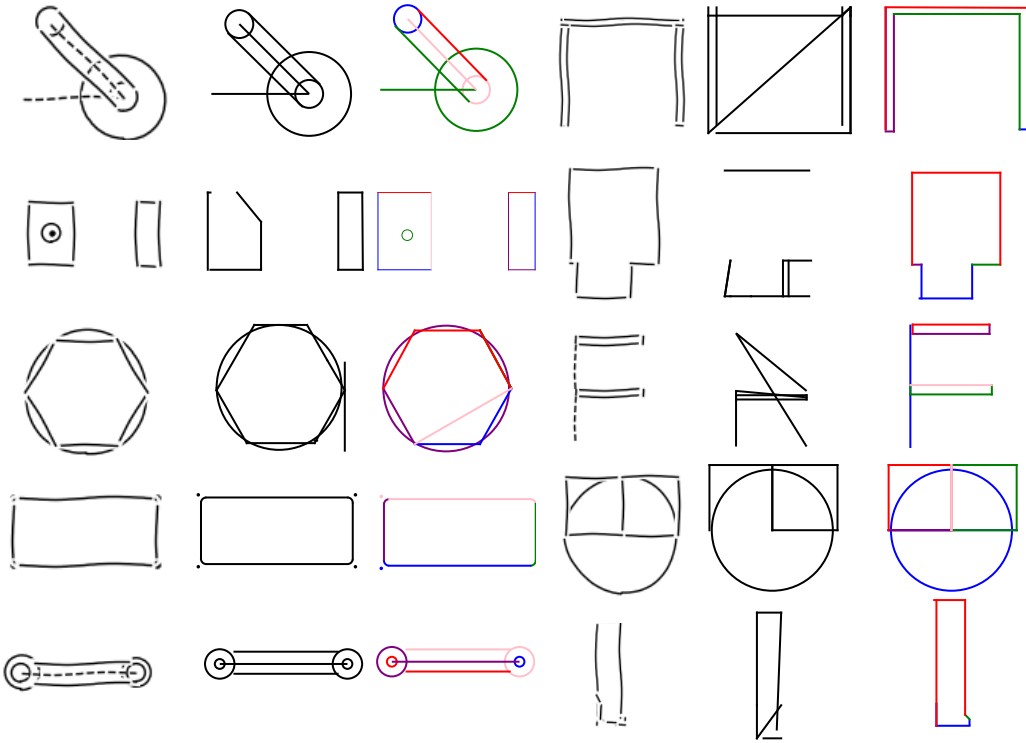

Figure 16: **Image conditioned generation**. Each example shows the input sketch image, the reconstruction result by autoregressive baseline, and our result.

| Config | Primitive | Constraint |
|---|---|---|
| Autoregressive | 0.575 | 0.301 |
| Ours | **0.711** | **0.368** |

Table 5: **Image-conditioned generation.** F-scores are reported for primitives and constraints.

We provide quantitative evaluation in Table 5 and visual comparison in Fig. 16, both showing that our model has superior performance than the autoregressive baseline, which again can be attributed to the more regular generation through sketch concept composition.

### A.9 Experiment on "CAD as Language" dataset

We also conduct preliminary experiments of our method on the dataset of [6], which comprises of millions of CAD sketches retrieved from OnShape [15]; in comparison, the SketchGraphs dataset [17] on which we have done the other experiments is similarly collected from OnShape but has a smaller scale. We filter the dataset by removing trivial or semantically ambiguous sketches and confine the sketch complexity such that the total number of primitives and constraints is within $[20, 90]$. In the end, we obtain about 2.5 million sketches and use 2.3 millions for training and the rest for testing. In comparison, there are about 1 million samples from SketchGraphs dataset used in the other experiments, where the maximum sketch graph size is 50 (Sec. 6).

To accommodate the increased complexity of this dataset, we increase the query number $k_{qry}$ to 6, the $\mathbb{L}^1$ library size to 15, and leave the rest hyperparameters unchanged. Examples of learned libraries and corresponding sketches are presented in Fig. 17. Examples of design intent parsing results are given in Fig. 18. We can see that our method obtains new modular concepts and parses more complex sketches; these results show that our framework works similarly on this new dataset.

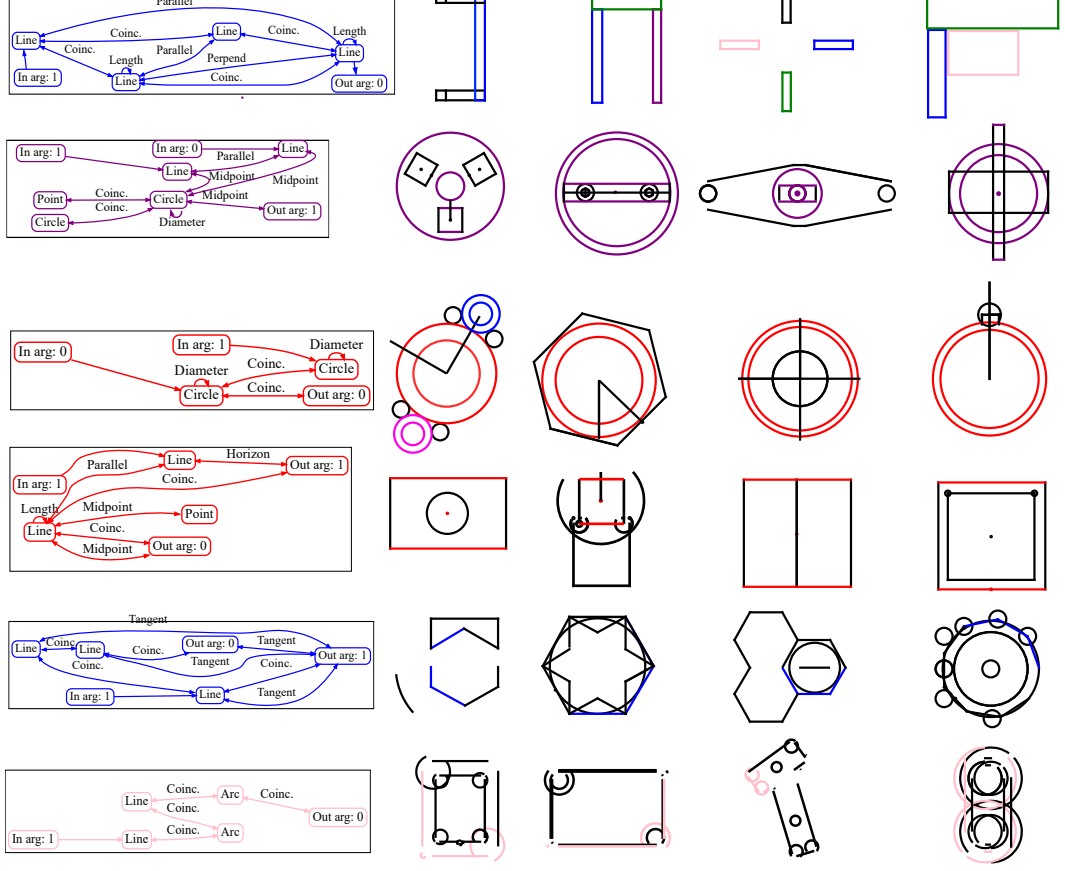

Figure 17: **Examples of library concepts learned from the CAD as Language dataset [6] and their corresponding sketch instances**. Different instances of the same library are highlighted in different colors.

## A.10 Broader impact

This work potentially improves the efficiency of CAD sketch design, which however does not replace other critical procedures of CAD. For example, the discovered concepts do not necessarily meet structure safety constraints, and should be subject to checking and validation procedures according to specific applications. The general methodology of program library induction presented in this work facilitates more structured and interpretable machine learning, which may enhance human and AI interaction but has no direct negative social impacts.

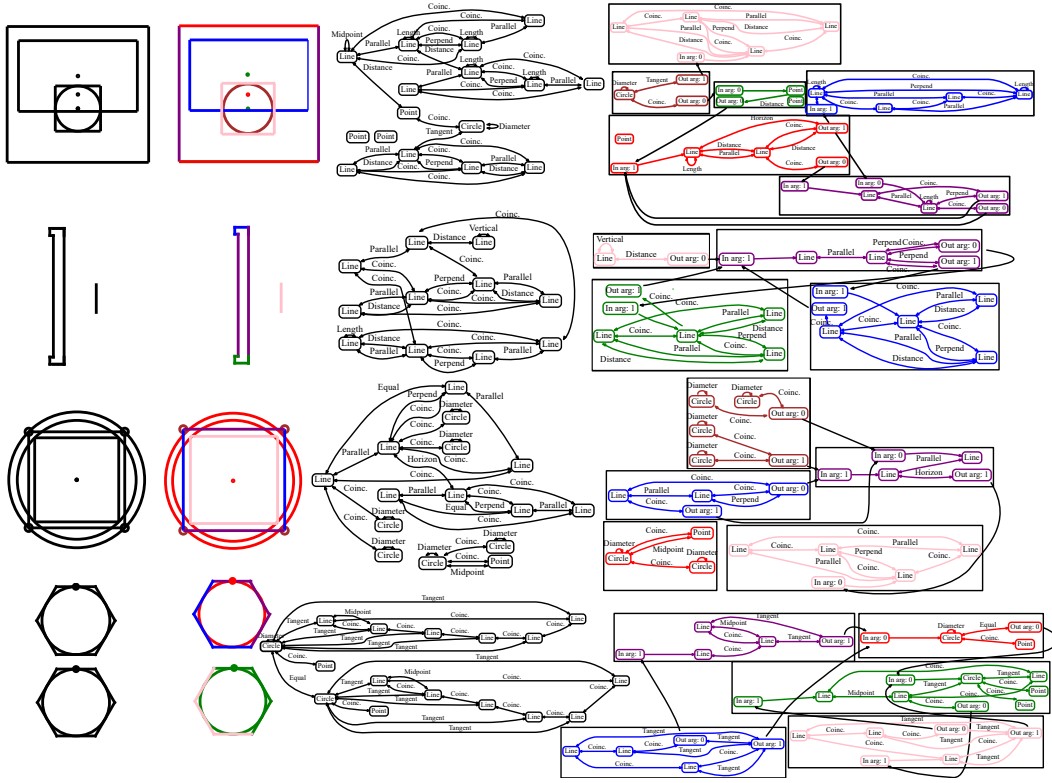

Figure 18: **Design intent parsing learned on the CAD as Language dataset [6]**. Each example shows the input raw sketch and corresponding constraint graph (in black), as well as our interpreted sketch made of modular concepts and corresponding modular constraint graph, where primitives and constraints are colored according to their encapsulating concepts.