# OpenReview forum: "Discovering Design Concepts for CAD Sketches"
_NeurIPS.cc/2022/Conference — NeurIPS 2022 Accept_

### Official Review · Reviewer_5G2f · 2022-07-04

**Rating:** 7
**Confidence:** 4
**Soundness:** 3 good
**Presentation:** 2 fair
**Contribution:** 4 excellent

**Summary:**

This paper proposes a novel learning based approach to discover the modular concepts (i.e., modular structure) from raw CAD sketches. To tackle the problem, the authors first define a domain specific language (DSL) such that modular concepts can be represented in a network-friendly manner.

A Trasnformer-based detection module takes in a CAD sketch sequence and outputs a a set of latent embeddings, which are further decoded to parameterized modular concepts by a generation module. The whole model is trained in an end-to-end self-supervised manner, using reconstruction loss plus regularization terms.

The authors perform experiments on a large scale CAD sketch dataset and mainly demonstrates its applications for design intent interpretation (i.e., parse modular concepts from a raw CAD sketch) and auto-completion (i.e., complete a partial CAD sketch).


**Questions:**

- Line 73-77, the description for $L^1$ composite type is confusing, probably because it's a abstract concept.    I can understand it conceptually but hard to match the mathematic formulation. If possible, I would suggest explicitly writing out the $L^1$ composite types in Figure 1 example such that readers can understand this concept more precisely.

- Line 134-135, "each constraint reference as a primitive index is directly embedded as a code", can you be more specific? How does a primitive index be embeded as a code? Is it via positional encoding?
- Line 184-185, "Collectively...R of shape $(2k_{qry}\cdot k_{L^0})\times(k_{qry}\cdot k_{L^0})$". How is the complete reference matrix R generated? From Line 168 and Line 175, in-concept and cross-concept reference matrices $(R_{T^1}, R_S)$ are generated separately. So is R just composed of $R_{T^1}$ and $R_S$?

- Figure 5(a), what is the dash line (on the left)?
- Sec. 5.2, what's the motivation for using a vector quantization loss? In other words, what's the benefit of building a discrete concept code space? Additionally, There is no ablation study on this loss term.


**Limitations:**

I think limitations are well discussed and I do not foresee any potential negative societal impact.

**Strengths And Weaknesses:**

Strengths:

- From the motivation side, I agree that CAD design intents, which are implicitly encoded in the CAD sketches, are best reflected in mid-level recurring patterns. It's an intellectually interesting problem whether one can discover these concepts from raw sketches. As perhaps a first attempt, this paper formulate this problem in a principled way by defining a domain specific language. It brings new knowledge to the field of ML-aided CAD design.
- The main technical contribution, from my point of view, is the concept representation that is based on the defined DSL. The proposed framework is built upon such representation and there are many interesting technical designs: 1) a DETR-like detection module in the context of CAD sketches; 2) the separation of topological structure and parameterization; 3) formulation of loss functions based on reference matrices.

Weakness:

- The design intent interpretation gives good results on some examples (Fig.3) but seem to fail in many cases (Fig. 9). For example, for those symmetric CAD sketches (e.g., Fig. 9 first row first column, sixth row first column, sixth row four column), the parsed decompositions do not respect the symmetry, even though they are relatively simple sketches. The question then is how often the method fails? It would be helpful to have a user study to see how much percentage of the results agree with human interpretation, but I understand the difficulty to do that.
- Auto completion is indeed an crucial feature in CAD design and I agree that the modular concepts should be helpful if they are used properly. Conceptually, a CAD auto completion method should be able to give multiple plausible suggestions for the user. However, this method is determinstic --- it only generates one completion results given the partial input. It doesn't seem easy to extend this work to allow multiple possible outputs, which limits its practical potentials.
- The presentation for the defined DSL could be improved. I would suggest explicitly writing out all the types for the Fig. 1 example such that readers can align the abstract concepts with graphical illustrations.

Overall, I think that the paper formulates an intellectually interesting problem in a princepled way and tries to solve it with many unique technical designs. Though the results are not good enough, making it still far from practical usage, I think they are acceptable for a first attempt in this particular problem. In addition, I do think its technical contributions are novel and beneficial to the community.

---

> ### Author Response · Authors · 2022-08-01
> **Response to other questions of Reviewer 5G2f**
>
> **Q1**: Failure cases in Fig.9.
>
> Since symmetry regularity is not explicitly formulated by the objectives of library induction learning, there is no guarantee that the discovered interpretations always observe symmetry. However, the objective of modularity should implicitly encourage symmetry to improve reuse. Indeed, this is empirically confirmed by the numerous sketches that have been interpreted with diverse degrees of symmetry despite lacking an explicit requirement, as shown in e.g. Figs.1,3,4,9,10, etc.
>
> Additionally in a future work, symmetry along with other expert designed regularity priors can be integrated into the learning objective, to discover concepts with stronger patterns.
>
>
> **Q2**: Multiple completions.
>
> We agree that providing multiple completions can be even more useful. Currently our framework only provides one plausible completion based on the training data distribution. In future work, we may introduce random sampling schemes to the decoding process, to allow for multiple completions. Such randomized decoding has been explored in works like [1,2].
>
> [1] Marjan Ghazvininejad, Omer Levy, Yinhan Liu, and Luke Zettlemoyer. 2019. Mask-Predict: Parallel Decoding of Conditional Masked Language Models. In EMNLP.
>
> [2] Huiwen Chang, Han Zhang, Lu Jiang, Ce Liu, William T. Freeman. 2022. MaskGIT: Masked Generative Image Transformer. In CVPR.
>
>
> **Q3**: Fig.1 types.
>
> Thanks for pointing this out. We have revised Fig.1 to include the complete DSL program representing the sketch graph.
>
> **Q4**: Index embedding.
>
> Yes, the index is embedded through a positional embedding layer that turns an integer into a latent code.
>
> **Q5**: $R$ matrix.
>
> $R$ contains two parts: for the reference entries inside a concept, they are directly copied from $R_{T^1}$; for the references across concepts, they are copied from $R_{cref}$ which is computed by multiplying $R_{T^1}$ and $R_S$ (line182).
>
> **Q6**: Dashed lines.
>
> These are construction lines, according to the convention of SketchGraphs.
>
> **Q7**: Motivation of vector quantization.
>
> The quantization of concept codes is necessary so that we can find a library of reusable concepts applicable for the whole dataset. Without this module, there are infinite variations of latent concept representations and no finite library can be obtained.

---

> > ### Comment · Reviewer_5G2f · 2022-08-09
> > **Reply**
> >
> > I appreciate the detailed response by the authors. I think all my questions are clarified and the revised Figure 1 makes the concept much more clear to me. No further questions from my side.

---

### Official Review · Reviewer_VMk7 · 2022-07-10

**Rating:** 6
**Confidence:** 4
**Soundness:** 3 good
**Presentation:** 3 good
**Contribution:** 3 good

**Summary:**

The authors proposed a method to decompose CAD sketches into sub modular concepts, for design intent parsing. Specifically, the task is treated as a problem of program library induction. The goal is to discover sketch concepts (modular structures) from raw sketches in the format of sketch graphs. DSL is formulated to enable a concise way to represent sketch concepts. There are two key factors to make concepts searching trainable, I.e., the dual representation and the parameterized structure for concepts. In particular, authors utilize a transformer-based object detection framework for implicitly detecting and encoding concepts. Then, a generation module is to explicitly generate shapes corresponding to the found concepts. Finally, a self-supervised reconstruction loss coupled with other two losses (concept quantization loss and modularity enhancement loss) are severed as objectives.

Experiments are conducted on a subset of SketchGraphs dataset. Reconstruction accuracy and sketch concept modularity are used for evaluation. The proposed method also is capable of auto-completion for CAD modeling.

**Questions:**

(1) I am curious if the discovered sketch concepts are actually some advanced primitive shapes used by other generative models for CAD sketches, such as [6] [13][17]?

(2) The input sketch is $L^0$ typed instance, i.e., CAD sketches. Could the proposed method possibly to be applied to free-hand sketches?

(3) How to learn the set of concept queries $[\bar{q}_i]$?

(4) How library $L^1$ is defined? And what is the impact of the library size which is 1000 currently?

(5) Is the output ( a set of sketch concepts) unique for an input sketch? Since different sets of basic shapes can compose the same raw sketch as well, and multiple answers might be all correct. Is that really safe to say the discovered sketch concepts are well-aligned to groundtruth design intents with a high reconstruction score and sketch concept modularity?

(6) Is there a scalability issue as I saw most of the cases shown in this paper produce three sketch concepts for a single sketch input?

**Ethics Review Area:**

["I don’t know"]

**Limitations:**

I doubt there is a scalability issue when the library size increases (1000 for the current setting), since transformer-based framework is used. As we know, the number of tokens has a significant impact on the computational cost.

**Strengths And Weaknesses:**

Strengths

- To parse raw sketch graphs into sketch concepts to reveal design intents is meaningful. This work proposed a method to explicitly discover sketch concepts, which seemed different from shape primitives used by previous works. This work is more like to segment sketches into semantic parts (sketch concepts).

- The proposed method can be trained in a self-supervised manner, which facilities easier network training.

- The paper is organized and well-writing, I really enjoy the paper reading.

Weaknesses

- The total loss is a bit complex, with multiple losses combined. And each wights are set empirically, which potentially requires efforts to fine-tune especially when the dataset is changed. This would lead prohibition in practical usage.

- The experiments are somehow limited. It would be better to provide more baseline methods for comparison if possible. In addition, the experimental analysis is limited too, especially about the design intent interpretation. So it is unclear how the discovered sketch concepts support design intent parsing.

---

> ### Author Response · Authors · 2022-08-01
> **Response to other questions of Reviewer VMk7**
>
> **Q1**: Loss weights tuning.
>
> Since the loss terms are all normalized by their corresponding set sizes, significant tuning of loss weights probably can be avoided when training on new datasets. We didn’t apply extensive parameter tuning to find these empirical weights, and found that the results are quite stable across a wide range of weight variations.
>
> **Q2**: Are the discovered concepts advanced primitives used by other generative models?
>
> Both our approach and the other generative models operate on the same set of primitives, i.e. points, lines, arcs, etc. and their constraints. Since the other generative models output such primitives one-by-one, it’s unclear what their advanced primitives are.
>
> **Q3**: Applying to free-hand sketches.
>
> Yes, our approach can be used to convert freehand sketches into vectorized and structured sketches. This is shown in A.8 of supplementary. Following the other generative models, we use the free-hand sketches as conditional input.
>
> **Q4**: Learning queries $[\overline{q}_i]$.
>
> These variables are trained end-to-end along with other network parameters.
>
> **Q5**: Definition of $\mathbb{L}^1$.
>
> We define the syntax of $\mathbb{L}^1$ concepts as shown in List 1 and learn the content of $\mathbb{L}^1$ library by end-to-end inductive learning.
>
> **Q6**: Different compositions for a sketch.
>
> Indeed as you have observed, there can be multiple compositions for a given raw sketch, so there is frequently no unique ground-truth design intent for a sketch. Our approach allows for discovering one possible decomposition for a sketch, guided by the necessary conditions of reconstruction and modularity. As can be seen from the results and noted by expert designers (see **Q1** of common questions), these discovered concepts are generally plausible.
>
> **Q7**: Three concepts per sketch.
>
> The maximum number of concepts per sketch is specified by $k_{qry}$. Most results shown in the paper are obtained with $k_{qry} = 5$. There are quite a few samples with 4 or 5 concepts (see Figs. 5, 9,10). A discussion of the impact of $k_{qry}$ on modularity and reconstruction quality is given in A.7 of supplementary, where we can see that a larger $k_{qry}$ leads to more scattered concepts with reduced modularity (see also Fig.14).
>
> **Q8**: Scalability with large library size.
>
> We note that the library is not directly consumed by transformer modules, but only used to provide quantized concept codes for a fixed number of queries per sample (i.e. $k_{qry}$). The quantization computation has linear complexity with respect to the library size. Therefore there is no scalability issue with library size.

---

### Official Review · Reviewer_Jko6 · 2022-07-11

**Rating:** 6
**Confidence:** 4
**Soundness:** 3 good
**Presentation:** 3 good
**Contribution:** 3 good

**Summary:**

This paper contributes an end-to-end multi-module architecture that learns design concepts from CAD sketches, a specific type of "sketches" that are composed with well-formed primitives and constraints to be used for further CAD modelling. The author(s) define a DSL to encode CAD sketches, and construct implicit and explicit representations of deep-learning based methods to detect design concepts and generate sketch graphs. The implicit and explicit representations respectively correspond to the two primary modules in the architecture, which first transforms raw sketches into learnt discrete latent representations (i.e., implicit), and subsequently a generation module takes these implicit representations and generates the final graph in an explicitly defined format.

This model is trained end-to-end on self-supervised objectives contributed by the author(s). The author(s) evaluated the models to have learnt reasonable design concepts and achieved greater auto-completing accuracy over a representative autoregressive baseline, on an established large-scale dataset of CAD sketches.


**Questions:**

Most of my questions are already embedded in the Weaknesses section above. To reiterate in more directly and clearly:

- Is this implicit / explicit representation framework entirely novel at the highest level, or are there other instantiations that already exist in other domains? (i.e., Is your contribution only applying this framework to CAD sketches, or did you contribute this framework in general?)
- Did a designer and/or domain expert review the learnt concepts formally or informally? If so, did they find them to be useful?
- Would you further clarify how the "removed primitives" were represented in the auto-completion task in the proposed method?
- How might this framework be applied to discover higher order concepts?

**Limitations:**

I think it is generally reasonable to consider the targeted task of CAD sketch modelling to not have significantly negative societal impacts as the author(s) have outlined. However, one might further consider the minor impact when these models are deployed in real CAD use-cases when the user might become over-reliant on trusting the model as an absolute source-of-truth. This can have varying degrees of negative consequences depending on the final application of the CAD models, and can become harmful such as when the generated CAD model is structurally unsafe.

**Strengths And Weaknesses:**

Strengths:
- Novel and well-designed architecture that both utilises latest advances in deep-learning and generates human-interpretable sketch graphs
- Detailed and well-defined DSL for encoding and interpreting CAD sketches
- Showed promising results in balancing reconstruction accuracy of graphs and ensuring modularity of outputs
- Achieved better performance on a significant auto-completing task over a representative baseline

Weaknesses:
- The greatest weakness of the current paper is the lack of explicit evaluation on the learnt design concepts (which is the primary claimed goal that the model should achieve). While I appreciate that the authors have shown extensive qualitative examples in the paper and supplementary materials, I am curious to see if a designer and/or domain expert find these learnt concepts to be useful as some of them are relatively trivial (e.g., rectangles).
- When comparatively evaluating autoregressive baselines and the proposed method on the auto-completing task, how were the "removed primitives" represented respectively? If this is done in the proposed method by replacing existing primitives with an additional 'mask' class while keeping other aspects of inputs unchanged, it might be slightly unfair to the autoregressive baseline. This is because the information of the overall length or some further details about the sketch to be auto-completed might be leaked through the masked input to the proposed method.
- The proposed method tackles a very specific task in an expert domain. However, I believe there could be greater applications for the proposed framework of implicit / explicit representation, which makes this a more minor weakness.
- Related to the above point on the lack of generality: the current description on the exact scope of novelty in the paper is not clear. Is this implicit / explicit representation framework entirely novel at the highest level, or are there instantiations of it that already exist in other domains? I also recommend the author(s) to list the specific contribution(s) at the end of the introduction.
- The current method only works at the first order, which the author(s) acknowledged. I wonder if they can further discuss how might the framework be applied (potentially recursively) to discover higher order concepts.

---

> ### Author Response · Authors · 2022-08-01
> **Response to other questions of Reviewer Jko6**
>
> **Q1**: How removed primitives are represented.
>
> These primitives are simply removed from the input sequence. Therefore neither our approach nor the baseline approach has any information of how long the target sequence should be. Indeed, the input to both methods is exactly the same.
>
> **Q2**: Scope of novelty.
>
> Thanks for the question. We don’t think that the implicit/explicit representation is novel at its most general level, as essentially all works that deal with explicit structures with neural networks have to transit from an implicit feature extraction stage to an explicit structure generation stage, as done in e.g. DETR and [1]. However, we believe that in terms of inductively learning declarative first-order programs as formulated by our DSL, the implicit-explicit representation and the structure-parameter separation have proved feasible through our work for the first time. Although we demonstrate the feasibility by CAD sketch concept learning, we believe that such programs are not restricted to CAD and can be interesting to a broader audience. We have mentioned this around line348. We will revise to make our contributions more clear in the introduction.
>
> [1] Tian, Y., Luo, A., Sun, X., Ellis, K., Freeman, W. T., Tenenbaum, J. B., & Wu, J. (2019). Learning to infer and execute 3d shape programs. ICLR.
>
> **Q3**: Extension to higher order.
>
> Our framework can potentially be extended to higher order induction through recursion. A general scheme has already been demonstrated in DreamCoder. More specifically within our setting, given a first-order library, we can process the entire dataset and replace each raw sketch with a composition of first-order library concepts, to obtain a new dataset whose elements are first-order concepts. Our framework can then be applied onto this new dataset to inductively learn second-order concepts that are composed of first-order concepts; applying this procedure recursively would build a hierarchy of concept libraries. We would like to explore this extension in future work.
>
> **Q4**: Limitations.
>
> Thanks for pointing this out. We agree that structural validity considerations are essential; our concept discovery facilitates design but does not replace other critical procedures in CAD and manufacturing. We will emphasize this point in the revision.

---

### Official Review · Reviewer_vFn4 · 2022-07-15

**Rating:** 8
**Confidence:** 4
**Soundness:** 4 excellent
**Presentation:** 3 good
**Contribution:** 4 excellent

**Summary:**

**Summary:**

The paper presents aims to present a method for decomposing a CAD sketch into a set of of _modular-concepts_ that are learned from a large corpus of CAD sketches. The modular concepts are compostions of lower-level primitives connected via constraints.

In order to do this, the authors design a DSL for expressing these modular-concepts, inspired by Lambda-Calculus. Then they use sequence encoder + vector quantized compression as a bottleneck to force the network to compress the sketch into a bunch of _concepts_. A sequence decoder takes concepts from  the concept library and instantiates them with parameters. The whole pipeline is trained end-to-end under a (permutation-equivariant) reconstruction loss, with permutation equivariance guaranteed by graph matching.

**Questions:**

- On P16, Fig. 8, subfig 4 from the top. We see the salmon colored concept which is basically one _Equal_ constraint. This seems sketchy. There are many other concepts that basically amount to a concept containing a single constraint. My first question is how many concepts were learned for this qualitative result, and could you please present a histogram of how many primitives/constraints each concept contains when training with different number of allowable concepts. It seems weird that the network would basically call an $\mathbb{L}^0$ type an $\mathbb{L}^1$. Does it have something to do with the data? I would like an explanation.
- The second question relates to the same subfigure - why do we have overlapping concepts - is this a drawing artifact or something significant?
- In List 1, what exactly is $b_{dash}$? I can assume it means whether to draw the line as dashed, but it would be much better to have an explanation.
- You mention type casting in P6, L214  but how exactly does that help you compare different types? Would be nice to have an example. Like if a prmitive is a line and the target is a circle, how exactly is the loss computed?
- In P5, L198, "decoding layer $dec_{param}(·)$ that is inverse of $enc_{param}(·)$ in Sec.4.1.". What exactly does inverse mean here? Do you convert logits to actual values by argmaxing and querying the unquantized value? Do you take a weighted sum of the logits? Please explain in more detail.
- Could you explain the 100% modularity in Table 1, when removing the sharpness loss? What exactly does it mean? As it is just a bunch of primitives ( constraint F1 score is too low), how exactly does the concept become 100% modular?
- (*Speculative*) This question more curiosity but would it be possible to run the method on the dataset from [1]? This dataset is a lot more complex than SketchGraphs, and it would be interesting to see the concepts that emerge from this more complex dataset.

**References:**

[1] https://github.com/deepmind/deepmind-research/tree/master/cadl

**Limitations:**

**Negative Social Impact**

I foresee no negative social impact from this work unless our Future Robot Overlords are unhappy with the concepts generated running this method.

**Technical Limitations**

I would like to delineate between the limitations of this paper versus that of the method.

_Paper:_

- The paper severely lacks good comparisons to other methods. While this is mainly because of a lack of code, it is still a limitation.
- The paper lacks comprehensive ablations - while the authors do a good job of ablating their contributions and how much they affect the modularity of their _concepts_, practioners might also want ablations relating to how many layers they need, and how to tune the VQ and how to rank concepts and so on. This does not hold me back from accepting the paper, but the paper might be improved if the authors add to the paper their negative results as well - eg that having too many codes in the VQ led to ..., or that too many layers never converged. I think adding such practical advice to the paper will make it more useful.

_Method:_

 - The limitations of this method would be having to retrain for every desired level of modularity.
 - Unless I am very wrong, I guess that the mathching part of the algorithm is somewhat unstable (The authors can correct me here). This is based on my experiences with DETR, and requires designing the cost matrices with considerable care.

**Strengths And Weaknesses:**

**Strengths:**

- The paper has a  very original idea - it uses previously existing building blocks like Transformers, Vector Quantization and Hungarian Matching for set-structured data - but in a creative way. I commend the authors on this.
- I found the paper very logically structured with a good introduction to the problem the authors are trying to set-up and solve. It clearly highlights how it is different from previous work [1,2], which relies on reconstruction from lower-level primitives (line, point, arc...).
I am not an expert in compilers/grammars/linguistics so I am not sure about the quality of the references in that domain, but they seem sufficient to me as an introduction to the state of the field, although they seem to a bit old judging from the years of the papers cited [3,4,5]. I cannot recite off the top of my head papers the authors might need to cite in order to get a better view of the current (~2022) landscape, but perhaps other reviewers could chime in here.
- The qualitative results are plenty and seem to support the paper well in terms of the diversity and quality  of the discovered concepts.
- The authors use the supplementary material very well and I think there are enough details to reimplement the paper, but perhaps not enough to reproduce the results.
- The qualitative figures are top-notch at showcasing what the method does.

**Weaknesses:**
- While I commended the authors on the general writing, some writing is very stilted. For starters, the authors do not use equation numbers in all places which makes it harder to reference equations.
- Continuing with equations, the notation seems a bit lacking - using $\mathbf{q}$, $\hat{\mathbf{q}}$,${\mathbf{q'}}$ makes it a bit harder to read the figures as well as the equations. Perhaps some cleaning up would be nice - as an example p-> primitives, e-> (contextualized) embeddings,  and you can retain q and q' for unquantized and quantized versions of the concepts respectively.
- Again continuing with equations, you mention indices implicitly often with "where the [index] iterates over ...". This often makes the sums harder to parse. It might be better to list out the indices explicitly. As an example, on L136, it is not really clear until ones reads the supplementary, what exactly $t.x$ runs over.
- There are not a lot of comparisons but that is mostly fine as there is not a lot of code from competitors.
- The authors do not discuss the weaknesses very well - to me a major weakness would be that a) the number of concepts per model is fixed b) the number of primitives per model is fixed and that to change both one needs to retrain the model. Additionally, the authors should mention explicitly that they do not handle constraints with parameters and engage in a preprocessing step. This is not a big deal as others simply drop them.  But I believe this should be mentioned explicitly. Currently it is a throwaway line in the supplementary.
- The authors do not describe their autoregressive baseline that well - how is the encoding performed - like the current submission or like SketchGen[1] or Vitruvion[2]? Please try and explain it in greater detail.

**Minor Weaknesses:**
- The paper would benefit by explictly stating that the constraints are described by a lambda calculus and adding a citation to an introductory book on the subject. Right now it is unclear as to why constraints have a $\lambda$-dot notation.



Overall, the paper is very original, has high quality, is reasonably clear and I hope is significant in the field.

**References:**

[1] Para W, Bhat S, Guerrero P, Kelly T, Mitra N, Guibas LJ, Wonka P. Sketchgen: Generating constrained cad sketches. Advances in Neural Information Processing Systems. 2021

[2] Seff, A., Zhou, W., Richardson, N., & Adams, R.P. (2021). Vitruvion: A Generative Model of Parametric CAD Sketches. ArXiv, abs/2109.14124.

[3] Kevin Ellis, Daniel Ritchie, Armando Solar-Lezama, and Josh Tenenbaum. Learning to infer graphics programs from hand-drawn images. In Advances in Neural Information Processing Systems, volume 31. Curran Associates, Inc., 2018.

[4] Kevin Ellis, Maxwell Nye, Yewen Pu, Felix Sosa, Josh Tenenbaum, and Armando Solar-Lezama. Write, execute, assess: Program synthesis with a repl. In Advances in Neural Information Processing Systems, volume 32. Curran Associates, Inc., 2019

[5]Lazar Valkov, Dipak Chaudhari, Akash Srivastava, Charles Sutton, and Swarat Chaudhuri. Houdini:Lifelong learning as program synthesis. In International Conference on Neural Information Processing Systems, 2018.

---

> ### Author Response · Authors · 2022-08-01
> **Response to other questions of Reviewer vFn4 (part 1)**
>
> **Q1**: Reproducing the results.
>
> We will release all code and data that can reproduce the results shown in the paper.
>
> **Q2**: Equation numbers, notations, index range.
>
> Thanks for the comments. We will revise for better readability.
>
> **Q3**: Discussion of weakness.
>
> Thanks for pointing this out. When new primitives or concepts need to be introduced, the whole model should at least be finetuned, if not retrained from scratch. Meanwhile, we note that a close variation of the vector quantization bottleneck can potentially avoid training the whole network for introducing new concepts, which is to use key-value pairs to localize the impact of new concepts and preserve learned ones, as shown in [1].
>
> We will highlight the handling of constraint parameters in the main text. Moreover, we note that these parameters can be included in the generation model without difficulty; we have chosen to skip them for the sake of design simplicity, as they can be reliably recovered from the generated primitive shapes.
>
> [1] Frederik Träuble, Anirudh Goyal, Nasim Rahaman, Michael Mozer, Kenji Kawaguchi, Yoshua Bengio, Bernhard Schölkopf. 2022. Discrete Key-Value Bottleneck. arXiv:2207.11240.
>
> **Q4**: Encoding of autoregressive baseline.
>
> As discussed in line311-312, we use the same encoding as ours for the baseline model for fair comparison, as the previous works have each used slightly different encodings.
>
> **Q5**: Citation for lambda calculus formulation.
>
> Thanks for pointing this out! We will add references to facilitate understanding.
>
> **Q6**: Fig.8, trivial $\mathbb{L}^1$ concept containing a single $\mathbb{L}^0$ typed element.
>
> Given a sketch, all the $\mathbb{L}^0$ elements are converted into encapsulating $\mathbb{L}^1$ typed concepts by our network. Therefore, an extra $\mathbb{L}^0$ element not fitting to any modular structure will be contained within a $\mathbb{L}^1$ concept to ensure reconstruction of the whole sketch. However, such *trivial* $\mathbb{L}^1$ types are not common. We have plotted the histogram of $\mathbb{L}^1$ type complexity in the revised draft (please see Fig.11(b) of supplementary), and find that there are 10 such $\mathbb{L}^1$ types out of the 1000 set.
>
> On the other hand, if we vary the number of allowable concepts per sketch, as shown in the experiment of changing $k_{qry}$ (A.7 of supplementary), we can see that modularity degrades with increased numbers of concepts. This however does not mean trivial $\mathbb{L}^1$ types abound, as e.g. when $k_{qry}=10$ the percentage of *trivial* $\mathbb{L}^1$ is 3.7%.
>
> **Q7**: Overlap in subfigure.
>
> This is a layout problem with the graphics software, and there should be no overlap. We have fixed it in the revision.
>
> **Q8**: $b_{dash}$.
>
> It means the line is a construction line and is drawn as dashed according to SketchGraphs convention.
>
> **Q9**: Type casting.
>
> As an example, a line primitive is matched to a circle target, then the parameter code of the primitive is decoded by $dec_{param}()$ into a 256-dim code (Fig.7), out of which we only take the segment corresponding to the target circle type. The segment encodes quantized circle properties and is compared with the target circle properties for loss computation.
>
> **Q10**: $dec_{param}()$.
>
> $dec_{param}()$ has a mirrored structure of $enc_{param}()$. It takes a latent parameter code as input and decodes it into a 256-dim code (Fig.7), which contains several segments corresponding to different primitive types. Each primitive property is represented by a 14-dim embedding code, from which a quantized property value is recovered by an inverse-embedding layer. During this inverse-embedding process, the logits are processed by argmax to query the quantized value. Following previous works (CAD-As-Language, SketchGen, Vitruvion), we always work with quantized attribute values as categorical variables during network training and inference.
>
> We will revise to make the decode process more explicit in the supplementary.
>
> **Q11**: 100% modularity in Table 1.
>
> We compute modularity as the percentage of in-concept references, out of *correctly reconstructed* constraints (line281-282). So Table 1 shows that, without the sharpness loss, very few constraints are properly reconstructed, and for these constraints the references are entirely within their encapsulating concepts, which is an expected result of the modularity enhancing bias loss (Sec.5.3).

---

> ### Author Response · Authors · 2022-08-01
> **Response to other questions of Reviewer vFn4 (part 2)**
>
> **Q12**: Test on the larger dataset.
>
> Thanks for the suggestion! We will try to run on this larger dataset with 5million sketches, while our results have been obtained on 1million sketches. We note that both datasets originate from OnShape user creations and probably have very similar distributions.
>
>
> **Q13**: Ablation on hyperparameters.
>
> Throughout our experiments, we find the hyperparameters $k_{qry}$ and $k_{arg}$ have a clear impact on modularity and reconstruction quality. The network layers do not impact much, except for extremely small networks that cannot learn well and too large networks that do not fit into our limited GPU memory.
>
> **Q14**: Unstable matching.
>
> We find that on convergence the matching is stable. In addition, the type casting and soft reference matrix all contribute to the insensitivity and robustness of our loss computation to matching variations. So we find the training processes are generally quite stable.

---

> > ### Comment · Reviewer_vFn4 · 2022-08-09
> > **Response to Rebuttal**
> >
> > I thank the authors for their efforts in responding to reviewer comments.
> >
> > The new submission reads better than the original with more explanations. I appreciate the new histograms for concept complexity. They do provide some new insights into the model.
> >
> > I already had a high score for the paper, so I won't change it.
> >
> > But I would still ask the authors to add equation numbers and improve the summation indexing. In addition, I would also like to see preliminary qualitative results on the DeepMind dataset.
> >
> > I disagree with the authors on the comment that the DeepMind and the SketchGen datasets have similar complexity. In my experience, the DeepMind dataset has more diversity and complexity. Therefore, the concepts learned there should be a lot more complex.
> >
> > I think the authors have responded in detail to the other reviewers as well, so I still recommend acceptance.

---

### Author Response · Authors · 2022-08-01
**Response to common questions from reviewers**

We thank the reviewers for the constructive and thought-provoking comments, and for the recognition of the novelty and contribution of this work. In this post we answer the common questions from several reviewers, and reply to the other questions separately. To support the discussion, we have updated the draft and supplementary with additional data and highlighted the changes; line numbers and figure/table indices in the discussion refer to the updated draft. Later we will revise the paper thoroughly based on all comments.

**Q1**: How expert designers assess the discovered concepts. (**Jko6**, **5G2f**)

We have informally discussed the question of modular sketch concept discovery and reviewed the results obtained by our method with expert designers. Their comments are generally positive. First, they note that in daily practice they would build their own libraries of reusable modular components, so that when new design requirements arrive they can quickly adapt and compose the components to fit for the changes. Second, they comment that such inductive modular concept learning over large scale dataset can be quite helpful for designers who serve a diverse range of customers with very different design requirements. Third, they note that the discovered components by our method are generally quite common for them, even though the complete sketches from SketchGraphs dataset containing those components can be unusual in terms of being incomplete or inexact, which is understandable as the SketchGraphs dataset collected from OnShape user creations has a lot practice drawings.

**Q2**: Limited comparison against baseline works. (**vFn4**, **VMk7**)

Up to now we still cannot find the official implementations of comparing autoregressive baselines. Therefore, we have implemented an autoregressive baseline following the works of SketchGen and Vitruvion, both of which consists of a primitive model and a constraint model pointing to the primitives. For fair comparison, we ensure the autoregressive baseline uses the same input sequence encoding as ours, and has the same scale of transformer layers as ours.


**Q3**: Impact of library size. (**vFn4**, **VMk7**)

In the revised draft, we provide the tests with different library sizes in Table 4 of supplementary, from which we see that the library size (i.e. VQ codebook length) should be sufficiently large to cover the concept variations in 1million sketches and allow for good reconstruction. Beyond that, extra library entries cover rare structures and have little impact on the overall results.

---

### Meta-Review · Area_Chair_5yxB · 2022-08-26

**Recommendation:** Accept
**Confidence:** Certain

**Metareview:**

As summarized by reviewer 5G2f, this paper proposes a novel learning-based approach to discover the modular concepts (i.e., modular structure) from raw CAD sketches. To tackle the problem, the authors first define a domain specific language (DSL) such that modular concepts can be represented in a network-friendly manner.

A Transformer-based detection module takes in a CAD sketch sequence and outputs a a set of latent embeddings, which are further decoded to parameterized modular concepts by a generation module. The whole model is trained in an end-to-end self-supervised manner, using reconstruction loss plus regularization terms.

The authors perform experiments on a large scale CAD sketch dataset and mainly demonstrate its applications for design intent interpretation (i.e., parse modular concepts from a raw CAD sketch) and auto-completion (i.e., complete a partial CAD sketch).

All reviewers recognize the novelty and contribution of this work, and the reviewer-author discussion was quite fruitful as many points, ranging from designer/user interaction, comparison to baseline methods, and issues with the library size are discussed and addressed. With such clear contribution and applicability to the CAD domain, I highly recommend the acceptance of this work.


**Award:**

No

---

### Decision · Program_Chairs · 2022-09-14

Accept